# IMAGE CLUSTERING CONDITIONED ON TEXT CRITERIA

**Sehyun Kwon**[†1]**, Jaeseung Park**[†1]**, Minkyu Kim**[◇]**, Jaewoong Cho**[◇]**, Ernest K. Ryu**[†*]**, Kangwook Lee**[◇♣*]
[†]Seoul National University, [◇]KRAFTON, [♣]University of Wisconsin–Madison, [*] Co-senior authors

## ABSTRACT

Classical clustering methods do not provide users with direct control of the clustering results, and the clustering results may not be consistent with the relevant criterion that a user has in mind. In this work, we present a new methodology for performing image clustering based on user-specified text criteria by leveraging modern vision-language models and large language models. We call our method **I**mage **C**lustering Conditioned on **T**ext **C**riteria (IC|TC), and it represents a different paradigm of image clustering. IC|TC requires a minimal and practical degree of human intervention and grants the user significant control over the clustering results in return. Our experiments show that IC|TC can effectively cluster images with various criteria, such as human action, physical location, or the person's mood, while significantly outperforming baselines.[2]

## 1 INTRODUCTION

Image clustering has been studied as a prototypical unsupervised learning task, and it has been used to organize large volumes of visual data (Platt et al., 2003), to reduce the cost of labeling an unlabeled image dataset (Russell et al., 2008; Schmarje et al., 2022), and to enhance image retrieval systems (Wu et al., 2000; Jégou and Chum, 2012). Modern deep image clustering methods are often evaluated against pre-defined class labels of datasets viewed as the ground truth.

In practice, however, a user may have a criterion in mind for how to cluster or organize a set of images. The user may even want multiple clustering results of the same dataset based on different criteria. (See Figure 1.) But, classical clustering methods offer no direct mechanism for the user to control the clustering criterion; the clustering criteria for existing methods are likely determined by the inductive biases of the neural networks and the loss function, data augmentations, and feature extractors used within the method. This necessitates a new paradigm in image clustering, enabling diverse outcomes from a single dataset based on user-specified criteria and revolutionizing the conventional, implicitly dictated clustering processes.

Recently, foundation models have received significant recent interest due to their ability to understand and follow human instructions at an unprecedented level. Large language models (LLMs) (Brown et al., 2020; Chowdhery et al., 2022; Touvron et al., 2023a;b; Chiang et al., 2023; OpenAI, 2023; Adams et al., 2023) perform remarkably well on a wide range of natural language tasks such as understanding, summarizing, and reasoning in zero- or few-shot settings. Vision-language models (VLMs) (Alayrac et al., 2022; Liu et al., 2023; Awadalla et al., 2023; Dai et al., 2023; Li et al., 2023a; Zhu et al., 2023; Gong et al., 2023) interpret natural language instructions in visual contexts and produce responses that seemingly exhibit in-depth image analyses and complex reasoning.

In this work, we present a new methodology based on foundation models for performing image clustering based on user-specified criteria provided in natural language text. We call our method **I**mage **C**lustering Conditioned on **T**ext **C**riteria (IC|TC), and it represents a different paradigm of image clustering: the user directs the method with the relevant clustering criterion, the same dataset can be clustered with multiple different criteria, and if the clustering results are not satisfactory, the user can edit the text criterion to iteratively refine the clustering results. IC|TC requires a minimal and practical degree of human intervention and grants the user significant control over the clustering results in return, and we argue that this makes IC|TC more practical and powerful compared to the classical purely unsupervised clustering methods.

---

[1] Work done at KRAFTON.    [2] Our code is available at https://github.com/sehyunkwon/ICTC.

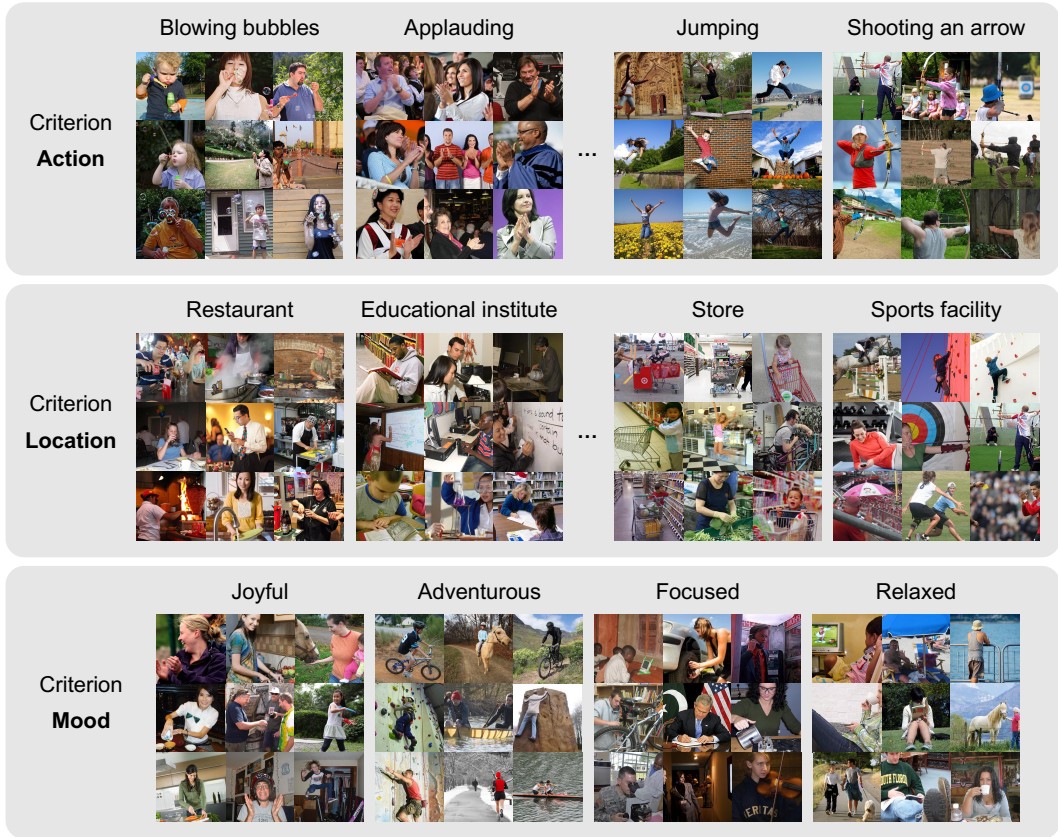

(a) Sample images from the clustering results on the Stanford 40 Action dataset. Each result is obtained using a different text criterion: `Action`, `Location`, and `Mood`.

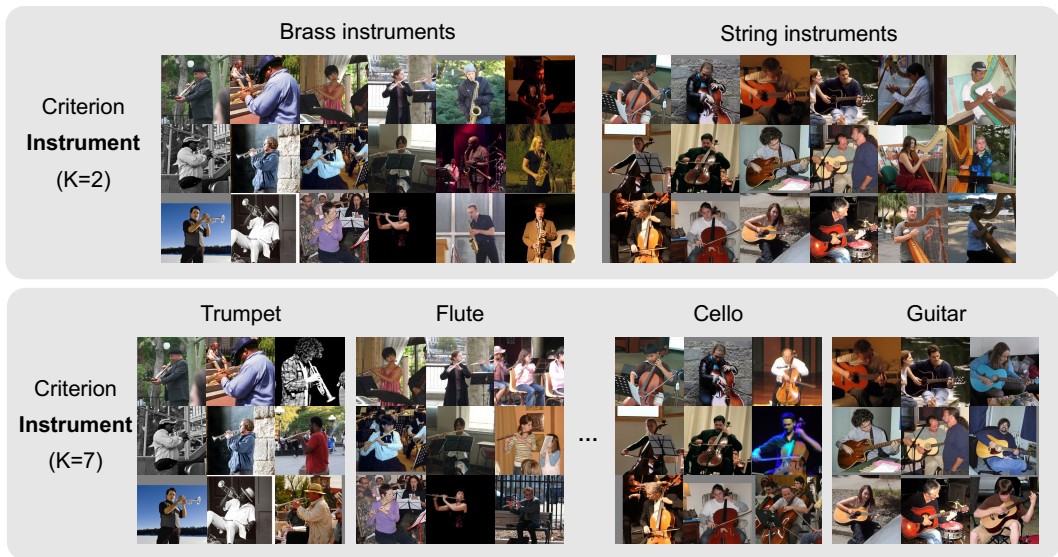

(b) Sample images from the clustering results on the PPMI dataset using the text criterion `Instrument` with different cluster numbers $K = 2$ and 7.

Figure 1: Sample images from clustering results of IC|TC. The method finds clusters consistent with the user-specified text criterion. Furthermore, IC|TC provides cluster names (texts above each image cluster) along with the clusters, enhancing the interpretability of clustering results.

## 1.1 CONTRIBUTION

Our main contributions are the proposal of the novel task of image clustering conditioned on text criteria and our method IC|TC for solving this task. The task is interesting because the setup where the user is willing and able to provide a textual description of the clustering criterion is practical, arguably more practical than the classical purely unsupervised clustering setup. The method IC|TC is interesting because it leverages modern multi-modal foundation models and solves the task well; our experiments demonstrate that IC|TC can indeed produce satisfactory clustering results consistent with the user-specified criteria.

## 2 TASK DEFINITION: IMAGE CLUSTERING CONDITIONED ON ITERATIVELY REFINED TEXT CRITERIA

The main task we consider in this work is defined as follows: Given a set of images, a number of clusters $K$, and a user-specified criterion expressed in natural language, partition the set of images into $K$ clusters such that the semantic meanings of the clusters are distinguished in a manner that is consistent with the specified user criterion.

Recent image clustering methods (Van Gansbeke et al., 2020; Park et al., 2021; Niu and Wang, 2021) find clusters that agree with pre-defined class labels for datasets such as CIFAR-10 ($\sim$90% accuracy). The semantic meanings of the clusters tend to correspond to the category of the foreground object, and the inductive biases of the neural networks and the loss function, data augmentations, and feature extractors used within the method are likely the cause of the clusters being chosen in this manner. In a given setup, however, the clusters returned by such classical clustering methods may not be consistent with the relevant criterion that a user has in mind.

**Iterative refinement of text criteria.** Under our main task, the text criterion is chosen through a process of iterative refinement: The user specifies a text criterion, performs clustering, examines the clustering results, and, if not satisfied, edits the text criterion to iteratively refine the clustering results. Sometimes, a user-defined text criterion immediately leads to a clustering result that is sufficiently consistent with what the user has in mind, but if not, this iterative prompt engineering procedure provides a practical means for converging to desired results. In practice, hyperparameters of classical clustering algorithms are chosen through an iterative process where the user inspects the clustering output and adjusts the parameters accordingly. In this work, we explicitly acknowledge the process of iteratively determining the text criterion and consider it to be part of the main task.

**Comparison with classical clustering.** Our task differs from classical clustering in that the user provides information characterizing the relevant criterion by which the images should be clustered. In contrast, classical clustering methods are purely unsupervised and use no such information.

Deep clustering methods are often evaluated against a pre-defined set of labels of a dataset, and such labels tend to focus on the type of object in the foreground. However, the question of how clustering algorithms could (or cannot) perform clustering with arbitrary criteria has been raised and studied in several prior works (Wolpert and Macready, 1997; Kleinberg, 2002; Caruana et al., 2006; Cui et al., 2007; von Luxburg et al., 2012; Caruana, 2013; McCarthy et al., 2020; Viswanathan et al., 2023). The use of user-defined text criteria makes our task not an instance of (classical) unsupervised clustering, but providing a text criterion is a necessary and practical intervention from the user if the goal is to perform clustering with arbitrary criteria.

**Comparison with zero-shot classification.** Our task differs from zero-shot classification in that zero-shot classification requires a pre-defined set of classes, and the goal is merely to assign images to these classes. In contrast, our task requires *both* finding the clusters and assigning images to the clusters. In fact, zero-shot classification can be considered an instance of our task when the user explicitly and precisely describes all $K$ clusters in the clustering criterion.

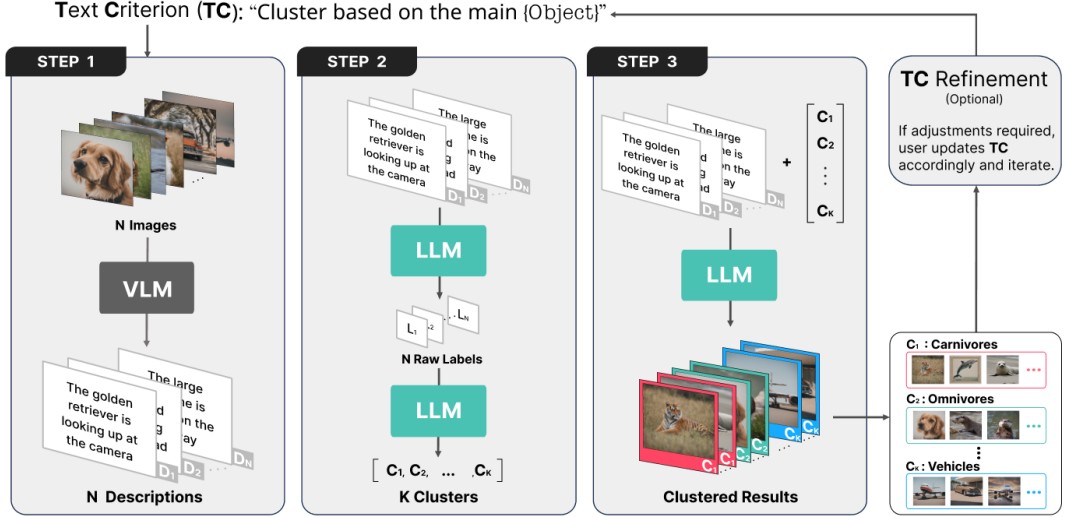

Figure 2: The IC|TC method. (Step 1) Vision-language model (VLM) extracts detailed relevant textual descriptions of images. (Step 2) Large language model (LLM) identifies the names of the clusters. (Step 3) LLM conducts clustering by assigning each description to the appropriate cluster. The entire procedure is guided by a user-specified text criterion (**TC**). (Optional **TC** Refinement). The user can update the text criterion if the clustering results are unsatisfactory. See Appendix B.4 for an unabridged sample output.

## 3 IC|TC: IMAGE CLUSTERING CONDITIONED ON TEXT CRITERIA

Our main method consists of 3 stages with an optional iterative outer loop. The user-specified text criterion **TC** is incorporated into 3 stages via text prompts roughly of the following form.

$P_{\text{step1}}(\mathbf{TC}) = $ `"Characterize the image using a well-detailed description"` $+ \mathbf{TC}$

$P_{\text{step2a}}(\mathbf{TC}) = $ `"Given a description of an image, label the image"` $+ \mathbf{TC}$

$P_{\text{step2b}}(\mathbf{TC}, N, K) = $ `"Given a list of {N} labels, cluster them into {K} words"` $+ \mathbf{TC}$

$P_{\text{step3}}(\mathbf{TC}) = $ `"Based on the image description,`
`            determine the most appropriate cluster"` $+ \mathbf{TC}$

The precise prompt for each experimental setup in this work is specified in Appendix B.3.1.

### 3.1 STEP 1: EXTRACT SALIENT FEATURES FROM THE IMAGE

In Step 1, the vision-language model (VLM) extracts salient features from the image in the form of text descriptions.

---

**Step 1** Vision-language model (VLM) extracts salient features

---

**Input:** Image Dataset $\mathcal{D}_{\text{img}}$, Text Criteria **TC**, Descriptions $\mathcal{D}_{\text{des}} \leftarrow []$
**Output:** $\mathcal{D}_{\text{des}}$
 1: **for** img in $\mathcal{D}_{\text{img}}$ **do**
 2:    $\mathcal{D}_{\text{des}}$.append( VLM(img, $P_{\text{step1}}(\mathbf{TC})$ )    `//append image description to` $\mathcal{D}_{\text{des}}$
 3: **end for**

---

The user's criterion **TC** determines the relevant features the VLM should focus on. For example, the user may wish to cluster with respect to the mood of a person in the image or the overall mood (atmosphere) of the scene. In such cases, the **TC** may slightly vary:

   `Criterion 1: Focus on the mood of the person in the center.`
   `Criterion 2: Describe the general mood by inspecting the background.`

## 3.2 STEP 2: OBTAINING CLUSTER NAMES

In Step 2, the large language model (LLM) discovers the cluster names in two sub-steps. In Step 2a, the LLM outputs raw initial labels of the images. Since the number of distinct initial labels is usually larger than $K$, in Step 2b, the LLM aggregates the raw initial labels into appropriate names of $K$ clusters. (Combining Steps 2a and 2b and asking the LLM to discover $K$ cluster names from $N$ image descriptions is infeasible due to the limited token lengths of the LLMs.)

---

**Step 2** Large Language Model (LLM) obtains $K$ cluster names

**Input:** Descriptions $\mathcal{D}_{\mathsf{des}}$, Text Criteria $\mathbf{TC}$, Dataset size $N$, Number of clusters $K$, $\mathcal{L}_{\mathsf{raw}} \leftarrow []$
**Output:** List of cluster names $\mathcal{C}_{\mathsf{name}}$
  1: **for** description in $\mathcal{D}_{\mathsf{des}}$ **do**
  2:    $\mathcal{L}_{\mathsf{raw}}$.append( LLM(description $+$ P$_{\mathsf{step2a}}(\mathbf{TC})$) )   //append raw label to $\mathcal{L}_{\mathsf{raw}}$
  3: **end for**
  4: $\mathcal{C}_{\mathsf{name}} =$ LLM($\mathcal{L}_{\mathsf{raw}}+$P$_{\mathsf{step2b}}(\mathbf{TC}, N, K)$)   //Step 2b can be further optimized

---

The simplest instance of Step 2b, described above, directly provides $\mathcal{L}_{\mathsf{raw}}$, the full list of raw labels. However, we find that it is more efficient to convert $\mathcal{L}_{\mathsf{raw}}$ to a dictionary with labels being the keys and numbers of occurrences of the labels being the values. When the same raw label occurs many times, this optimization significantly reduces the token length of the input to the LLM of Step 2b.

Careful prompt engineering of P$_{\mathsf{step2b}}(\mathbf{TC}, N, K)$ allows the user to refine the clusters to be consistent with the user's criteria. For example, the user may append additional text prompts such as:

```
When categorizing the classes, consider the following criteria:
1. Merge similar clusters. For example, [sparrow, eagle, falcon,
     owl, hawk] should be combined into 'birds of prey.'
2. Clusters should be differentiated based on the animal's habitat.
```

## 3.3 STEP 3: CLUSTERING BY ASSIGNING IMAGES

In Step 3, images are assigned to one of the final $K$ clusters. The text criterion $\mathbf{TC}$, text description of the images from Step 1, and the $K$ cluster names from Step 2 are provided to the LLM.

---

**Step 3** Large Language Model (LLM) assigns clusters to images

**Input:** Descriptions $\mathcal{D}_{\mathsf{des}}$, Text Criteria $\mathbf{TC}$, List of cluster names $\mathcal{C}_{\mathsf{name}}$, RESULT$\leftarrow[]$
**Output:** RESULT
  1: **for** description in $\mathcal{D}_{\mathsf{des}}$ **do**
  2:    RESULT.append( LLM(description$+$P$_{\mathsf{step3}}(\mathbf{TC})$))   //append assigned cluster
  3: **end for**

---

## 3.4 ITERATIVELY EDITING THE ALGORITHM THROUGH TEXT PROMPT ENGINEERING

---

**Main method** IC|TC

**Input:** Dataset $\mathcal{D}_{\mathsf{img}}$, Text Criteria $\mathbf{TC}$, ADJUST $\leftarrow$ True
  1: **while** ADJUST **do**
  2:    RESULT $\leftarrow$ **do Steps 1–3** conditioned on $\mathbf{TC}$
  3:    **if** User determines RESULT satisfactory **then**
  4:       ADJUST $\leftarrow$ False
  5:    **else**
  6:       $\mathbf{TC} \leftarrow$ Update $\mathbf{TC}$   //user writes updated $\mathbf{TC}$
  7:    **end if**
  8: **end while**

---

Our main method IC|TC is described above. Upon performing the clustering once, if the clusters are not sufficiently consistent with the specified text criterion $\mathbf{TC}$ or if the $\mathbf{TC}$ turns out to not precisely specify what the user had in mind, the user can update the $\mathbf{TC}$. This iterative process may continue until the clustering result is satisfactory, as judged by the user.

Table 1: Clustering with varying text criteria. Accuracies labeled with * are evaluated by having a human provide ground truth labels for 1000 randomly sampled images. In this experiment, we used LLaVA for VLM and GPT-4 for LLM.

| Dataset | Criterion | SCAN | Ours |
|---|---|---|---|
| Stanford 40 Action | Action | 0.397 | **0.774** |
| | Location | 0.359* | **0.822*** |
| | Mood | 0.250* | **0.793*** |
| PPMI | M.I. (K=7) | 0.632 | **0.964** |
| | M.I. (K=2) | 0.850 | **0.977** |
| | Location (K=2) | 0.512 | **0.914** |
| CIFAR-10-Gen | Object | **0.989** | 0.987 |

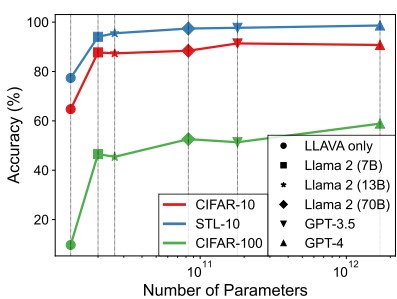

Figure 3: Effect of LLM selection.

## 3.5 PRODUCING CLUSTER LABELS

Classically, the unsupervised clustering task does not require the method to produce labels or descriptions of the output clusters. Notably, however, IC|TC produces names describing the clusters. This is a significant advantage of IC|TC as it makes the clustering results more directly and immediately interpretable.

## 4 EXPERIMENTS

We now present experimental results demonstrating the effectiveness of IC|TC. In this section, we partially describe the settings and results while deferring much of the details to the appendix. In particular, the precise text prompts used can be found in Appendix B.3.1.

IC|TC crucially relies on the use of foundation models, specifically a vision-language model (VLM) and a large language model (LLM) that have undergone instruction tuning. In our experiments, we mainly use LLaVA (Liu et al., 2023) for the VLM and GPT-4 (OpenAI, 2023) for the LLM, but Section 4.5 and Appendix B.2 presents ablation studies investigating how the performance is affected when other foundation models are used.

## 4.1 CLUSTERING WITH VARYING TEXT CRITERIA

In this experiment, we show that varying the text criterion **TC** indeed leads to varying clustering results of a single image dataset. The results demonstrate that IC|TC is highly flexible and can accommodate a variety of text criteria.

We use the Stanford 40 Action Dataset (Yao et al., 2011), which contains 9,532 images of humans performing various actions. The dataset comes with image labels describing a subject's action among 40 classes, such as reading, phoning, blowing bubbles, playing violin, etc. We additionally define two different collections of labels. The first collection contains 10 classes describing the location, such as restaurant, store, sports facility, etc. The second collection contains 4 classes describing the mood of the scene, specifically joyful, adventurous, relaxed, and focused.

We utilize three text criteria, Action, Location, and Mood, to obtain three distinct clustering results. We evaluate the results based on how accurately the methods recover the three collections of labels described previously. This degree of control would be difficult or impossible for classical deep clustering methods. We compare our results against the prior deep clustering method SCAN (Van Gansbeke et al., 2020) and present the results in Table 1. Image samples are in Figure 1a.

(Note that we do not have the 9,532 ground truth labels for the Location and Mood criteria. Therefore, we evaluate accuracy by having a human provide ground truth labels on 1000 randomly sampled images.)

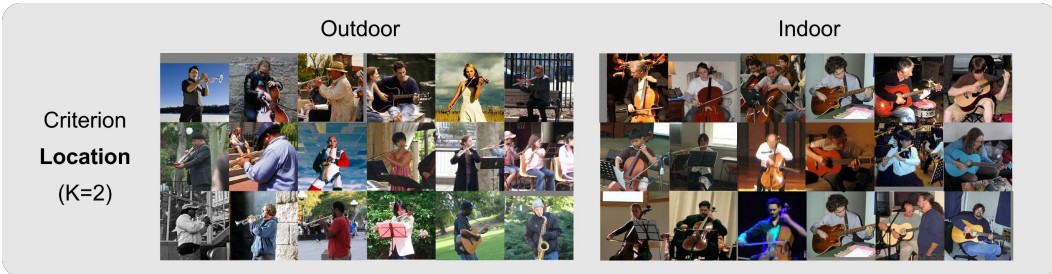

Figure 4: Sample images from the clustering results on the PPMI dataset using text criterion `Location` and cluster number $K = 2$.

## 4.2 CLUSTERING WITH VARYING GRANULARITY

In this experiment, we show that IC|TC can automatically control the granularity of clustering results by adjusting $K$, the number of clusters. We find that the cluster descriptions returned by IC|TC are highly interpretable and that the images are assigned to the clusters well for various values of $K$.

We use the People Playing Musical Instrument (PPMI) dataset (Wang et al., 2010; Yao and Fei-Fei, 2010), which contains 1,200 images of humans interacting with 12 different musical instruments. We select 700 images across 7 classes from the original dataset to reduce the size and difficulty of the task.

We use the text criterion `Musical Instrument` with number of clusters $K = 2$ and $K = 7$. With $K = 7$, images are indeed grouped into clusters such as violin, guitar, and other specific instruments, and 96.4% accuracy against the ground truth label of PPMI is achieved. With $K = 2$, images are divided into 2 clusters of brass instrument and string instrument and achieve a 97.7% accuracy. To clarify, we did not specifically instruct IC|TC to group the 7 instruments into brass and string instruments; the hierarchical grouping was discovered by IC|TC.

As an additional experiment, we also cluster the same set of images with the text criterion `Location` and $K = 2$. In this case, the images are divided into 2 clusters of indoor and outdoor, and achieve a 91.4% accuracy. We again compare our results against SCAN (Van Gansbeke et al., 2020) and present the results in Table 1. Image samples are provided in Figure 4.

## 4.3 COMPARISON WITH CLASSICAL CLUSTERING METHODS

In this experiment, we compare IC|TC against several classical clustering algorithms on CIFAR-10, STL-10, and CIFAR-100. The three datasets have 10, 10, and 20 classes and 10,000, 8,000, and 10,000 images, respectively. We use the text criterion `Object` with the number of clusters equal to the number of classes in the dataset. The results in Table 2 show that IC|TC significantly outperforms classical clustering methods on CIFAR-10, STL-10 and CIFAR-100. Clustered sample images are provided in Appendix B.6.

This comparison is arguably unfair against the classical clustering methods as they do not utilize foundation models or any pre-trained weights. Nevertheless, our results demonstrate that IC|TC is competitive when the goal is to cluster images based on the foreground object type.

## 4.4 FAIR CLUSTERING THROUGH TEXT CRITERION REFINEMENT

Existing clustering methods sometimes exhibit biased results, and measures to mitigate such biases have been studied (Li et al., 2020; Zeng et al., 2023). Since foundation models are known to learn biases in their training data (Bommasani et al., 2022), IC|TC has the risk of propagating such biases into the clustering results. In this experiment, we show that by simply adding a prompt along the line of `"Do not consider gender"` to the text criterion, we can effectively mitigate biases in the clustering results.

FACET (Gustafson et al., 2023) is a benchmark dataset for evaluating the robustness and algorithmic fairness of AI and machine-learning vision models. It comprises 32,000 diverse images labeled with

Table 2: Comparison with classical clustering methods using criterion `Object`. IC|TC outperforms state-of-the-art methods on CIFAR-10, STL-10 and CIFAR-100.

| Method | CIFAR-10 | | | STL-10 | | | CIFAR-100 | | |
|---|---|---|---|---|---|---|---|---|---|
| | ACC ↑ | NMI ↑ | ARI ↑ | ACC ↑ | NMI ↑ | ARI ↑ | ACC ↑ | NMI ↑ | ARI ↑ |
| IIC (Ji et al. (2019)) | 0.617 | 0.511 | 0.411 | 0.596 | N/A | N/A | 0.257 | N/A | N/A |
| SCAN (Van Gansbeke et al. (2020)) | 0.883 | 0.797 | 0.772 | 0.809 | 0.698 | 0.646 | 0.507 | 0.468 | 0.301 |
| SPICE (Niu and Wang (2021)) | 0.926 | 0.865 | 0.852 | 0.938 | 0.872 | 0.870 | 0.584 | 0.583 | 0.422 |
| RUC (Park et al. (2021)) | 0.903 | N/A | N/A | 0.867 | N/A | N/A | 0.543 | N/A | N/A |
| TCL (Yunfan et al. (2022)) | 0.887 | 0.819 | 0.780 | 0.868 | 0.799 | 0.757 | 0.531 | 0.529 | 0.357 |
| LLaVA only | 0.647 | 0.455 | 0.442 | 0.774 | 0.587 | 0.589 | 0.097 | 0.022 | 0.014 |
| Ours (LLaVA + Llama 2) | 0.884 | 0.789 | 0.759 | 0.974 | 0.939 | 0.944 | 0.526 | 0.554 | 0.374 |
| Ours (BLIP-2 + GPT-4) | **0.975** | **0.941** | **0.947** | **0.993** | **0.982** | **0.985** | 0.584 | **0.690** | **0.429** |
| Ours (LLaVA + GPT-4) | 0.910 | 0.823 | 0.815 | 0.986 | 0.966 | 0.970 | **0.589** | 0.642 | 0.422 |

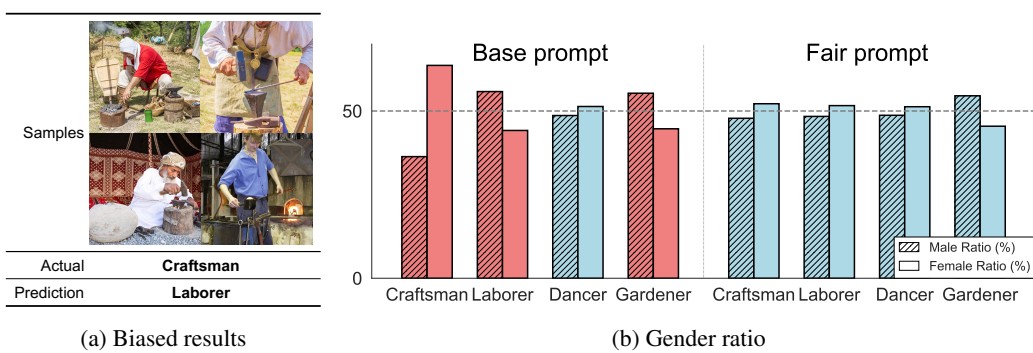

(a) Biased results           (b) Gender ratio

Figure 5: (a) Biased results showing that male 'Craftsman' tend to be misclassified as 'Laborer'. (b) Gender ratio of each cluster. When the ratio between males and females differs by more than 10%, the bar is colored red. Bias is mitigated by refining the text criterion into a 'Fair prompt'.

several attributes, including 52 occupation classes. For this experiment, we sampled 20 images each for men and women from the craftsman, laborer, dancer, and gardener occupation classes, 160 images in total.

For this experiment, we define fairness to be achieved when each cluster maintains an equal proportion of genders. When we use the text criterion `Occupation`, IC|TC exhibited a gender bias. To mitigate this bias, we introduced a simple negative prompt, instructing IC|TC to not take gender into consideration and instead to focus on the activity. When the clustering was repeated, the results were promising: the gender ratio disparities in the craftsman and laborer clusters improved by $27.2\% \rightarrow 4.4\%$ and $11.6\% \rightarrow 3.2\%$, respectively. Furthermore, the Dancer and Gardener clusters also experienced marginal reductions in disparities by $2.8\% \rightarrow 2.6\%$ and $10.6\% \rightarrow 9.0\%$, respectively. The results are shown in Figure 5.

## 4.5 FURTHER ANALYSES

**Ablation studies of LLMs and VLMs.** We conduct an ablation study to evaluate whether LLMs actually serve a significant role in our methodology since one may wonder whether the vision-language model (VLMs) alone is sufficient. When we perform a 'LLaVA only' experiment that does not utilize an LLM, the performance is considerably lower. However, when we use LLMs of varying sizes, the performance is not affected significantly. The results and details are provided in Figure 3 and Appendix A.2. The results lead us to conclude that the LLM serves a crucial role (the VLM by itself is not sufficient), but the size of the LLM does not seem to be very important.

We also fix the LLM to GPT-4 and perform an ablation study on the choice of vision-language model (VLM). As an image captioning model, ClipCap (Mokady et al., 2021) cannot perform text conditioning, and this leads to poor performance. Blip-2 (Li et al., 2023b) and LLaVA (Liu et al., 2023) can extract information relevant to the text criteria, and they exhibit strong strong performance. The results and details are provided in Appendix A.1.

**Data Contamination.** When evaluating research using foundation models, the potential of data contamination is a significant concern (Wei et al., 2022; Du et al., 2022). The datasets we use to measure accuracy, namely CIFAR10, STL-10, CIFAR-100, and Stanford 40 Action, may have been used in the training of LLaVA. If so, the validity of the accuracy measurements comes into question.

To address this concern, we conducted an experiment with synthetically generated images. Specifically, we use Stable Diffusion XL (Rombach et al., 2022) and the CIFAR-10 labels to generate 1000 CIFAR-10-like images, and we call this dataset CIFAR-10-Gen. See Appendix B for further details. On this synthetic data, IC|TC achieves 98.7% accuracy. The fact that the accuracy on CIFAR-10-Gen is no worse than the accuracy on the actual CIFAR-10 dataset gives us confidence that the strong performance of IC|TC is likely not due to data contamination.

(Strictly speaking, the training data for Stable Diffusion may contain the CIFAR-10 images, and if so, we are not completely free from the risk of data contamination. However, the CIFAR-10-Gen dataset does not seem to contain exact copies of CIFAR-10 images, and we argue that the synthetic generation significantly mitigates the risk of data contamination.)

## 5 RELATED WORK

**Image clustering.** Modern deep clustering methods (Van Gansbeke et al., 2020; Park et al., 2021; Niu and Wang, 2021; Yunfan et al., 2022) adopt a multi-stage training approach. They begin with representation learning, which finds a representation that maps similar images to similar features, and then perform unsupervised clustering based on these feature representations. Additionally, to obtain more meaningful semantics, Zhong et al. (2021); Shen et al. (2021) proposed contrastive learning at not only the instance level but also at the cluster level. Misra and Maaten (2020); Cho et al. (2021); Kwon et al. (2023); Long et al. (2023); Metaxas et al. (2023) proposed specially designed representation learning for certain clustering criteria. The concurrent work Li et al. (2023c) is particularly relevant to our work as it presents Text-Aided Clustering (TAC), which leverages text as external knowledge to enhance image clustering performance. Specifically, Li et al. (2023c) enhanced feature discriminability by selecting specific WordNet nouns of images and mutually distilled the neighborhood information between the text and image modalities.

**Foundation models.** In recent years, foundation models have been improving at a remarkable pace, and combined with instruction tuning (Sanh et al., 2022; Ouyang et al., 2022; Wei et al., 2022), these foundation models can be applied more flexibly to downstream tasks. Vision-Language Models (VLMs) (Alayrac et al., 2022; Liu et al., 2023; Awadalla et al., 2023; Dai et al., 2023; Li et al., 2023a; Zhu et al., 2023; Gong et al., 2023) can provide users with appropriate descriptions of given images according to the requirements of the input prompt. Large language models (LLMs) (Chowdhery et al., 2022; Touvron et al., 2023a;b; OpenAI, 2023) exhibit remarkable abilities in a wide range of natural language processing tasks such as text summarization. Recently, Radford et al. (2021); Jia et al. (2021); Li et al. (2022); Dinh et al. (2022); Geng et al. (2023); Menon and Vondrick (2023); Zhang et al. (2022); Cai et al. (2023); Ren et al. (2023) have shown computer vision problems with no direct connection to language can be successfully addressed using large language models.

**Image retrieval.** Image retrieval aims to find images from a database that are relevant to a given query. This crucially differs from clustering in that clustering requires both finding the clusters and assigning the images to them; image retrieval techniques are very relevant to the sub-task of cluster assignment but not to the sub-task of finding the clusters. The fundamental approach in image retrieval is to assess the similarity among image features. Current approaches focus on two kinds of image representations: global features and local features. For global representations, Babenko et al. (2014); Tolias et al. (2015); Gordo et al. (2016); Cao et al. (2020); Lee et al. (2023) extracts activations from deep CNNs and aggregates them for obtaining global features. For local representations, Yi et al. (2016); Noh et al. (2017); Vassileios Balntas and Mikolajczyk (2016); DeTone et al. (2018); He et al. (2018); Dusmanu et al. (2019); Revaud et al. (2019) proposed well-embedded representations for all regions of interest. Recent state-of-the-art methods (Noh et al., 2017; Simeoni et al., 2019; Cao et al., 2020; Zhang et al., 2023; Wu et al., 2023) typically followed a two-stage paradigm: initially, candidates are retrieved using global features, and then they are re-ranked with local features. Recently, Vo et al. (2019); Liu et al. (2021); Baldrati et al. (2022); Tian et al. (2023) proposed to condition retrieval on user-specified language.

## ACKNOWLEDGMENTS

EKR was supported by the National Research Foundation of Korea (NRF) Grant funded by the Korean Government (MSIP) [NRF-2022R1C1C1010010] and the Creative-Pioneering Researchers Program through Seoul National University. SK and EKR were partly supported by the Institute of Information & communications Technology Planning & Evaluation (IITP) grant funded by the Korean government (MSIT) [NO.2021-0-01343-004, Artificial Intelligence Graduate School Program (Seoul National University)]. We thank Dimitris Papailiopoulos and Yong Jae Lee for providing insightful discussion. We thank Byeong-Uk Lee for providing valuable feedback on the manuscript.

## ETHICS STATEMENT

Our methodology provides users with direct control over the clustering results, but this agency could be used maliciously to produce unfair and discriminatory results. However, it is unlikely that our work will be responsible for new unfair results that could not already be produced with a malicious user's direct and overt intervention. On the other hand, it is possible for biases already in foundation models to propagate into our clustering methodology. Section 4.4 explicitly discusses this possibility and offers measures to mitigate such biases, and a well-intentioned user following the guidance of Section 4.4 is unlikely to amplify biases in the foundation models through the use of our method.

## REPRODUCIBILITY STATEMENT

In this work, we use publically available datasets, describe the methodology in precise detail, and make our code available at `https://github.com/sehyunkwon/ICTC`. Of the two main foundation models we use, the vision-language model LLaVA (Liu et al., 2023) is fully open-source. However, the large language model GPT-4 (OpenAI, 2023) is a proprietary model, and we accessed it through the API offered by OpenAI. The API cost to conduct the experiments presented in this work was less than $3,000 (USD), so we argue that the proprietary API cost does not pose a significant barrier in terms of reproducibility. However, if OpenAI were to discontinue access to the GPT-4 version that we used, namely `api-version=2023-03-15-preview`, or if OpenAI discontinues access to GPT-4 altogether, then our experiments will no longer be exactly reproducible.

To address this concern, we carry out an ablation study that uses the open-source large language model Llama 2 (Touvron et al., 2023b) and observe that a similar, albeit slight worse, performance is attained. See Figure 3 and Appendix A.2. Therefore, even if GPT-4 becomes unavailable in the future, the results of this work will be similarly reproducible by using Llama 2 or any other large language model of power comparable to or stronger than Llama 2 and GPT-4.

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

# A ABLATION STUDIES

## A.1 ABLATION STUDY OF VISION-LANGUAGE MODELS

We fix the LLM to GPT-4 and perform an ablation study on the choice of vision-language model (VLM), specifically ClipCap (Mokady et al., 2021), Blip-2 (Li et al., 2023b), and LLaVA (Liu et al., 2023), during Step 1. ClipCap is unable to take in a text prompt and therefore the text criterion is not reflected in the resulting caption. Blip-2 and LLaVA can perform instructed zero-shot image-to-text generation and demonstrate the ability to interpret natural language instructions within visual contexts, producing responses that suggest thorough image analyses in our setting. This capability allows them to be used effectively within IC|TC, and we expect that any recent VLMs can likewise be utilized. Results are presented in Tables 3, 4, and 5.

Table 3: Ablation study of VLMs in CIFAR-10, STL-10 and CIFAR-100 datasets when using clustering criterion as `Object`.

| Method | CIFAR-10 | | | STL-10 | | | CIFAR-100 | | |
|---|---|---|---|---|---|---|---|---|---|
| | ACC ↑ | NMI ↑ | ARI ↑ | ACC ↑ | NMI ↑ | ARI ↑ | ACC ↑ | NMI ↑ | ARI ↑ |
| ClipCap | 0.636 | 0.605 | 0.524 | 0.722 | 0.729 | 0.647 | 0.365 | 0.396 | 0.214 |
| BLIP-2 | **0.975** | **0.941** | **0.947** | **0.993** | **0.982** | **0.985** | 0.584 | **0.690** | **0.429** |
| LLaVA | 0.910 | 0.823 | 0.815 | 0.986 | 0.966 | 0.970 | **0.589** | 0.642 | 0.422 |

Table 4: Ablation study of VLMs with Stanford 40 Actions dataset and clustering criteria `Action`, `Location` and `Mood`. N/W (Not Working) means: In Step 3, LLM responds that there is no appropriate label to assign the description obtained in Step 1. Accuracies labeled * are evaluated by having a human provide ground truth labels on 1000 randomly sampled images.

| Method | Action | | | Location | | | Mood | | |
|---|---|---|---|---|---|---|---|---|---|
| | ACC ↑ | NMI ↑ | ARI ↑ | ACC ↑ | NMI ↑ | ARI ↑ | ACC ↑ | NMI ↑ | ARI ↑ |
| ClipCap | 0.250 | 0.374 | 0.086 | 0.293* | 0.282* | 0.167* | 0.377* | 0.098* | 0.057* |
| BLIP-2 | 0.427 | 0.621 | 0.335 | 0.483* | 0.415* | 0.319* | 0.286* | 0.009* | 0.005* |
| LLaVA | **0.774** | **0.848** | **0.718** | **0.822*** | **0.695*** | **0.669*** | **0.793*** | **0.512*** | **0.525*** |

Table 5: Ablation study of VLMs in People Playing Musical Instrument (PPMI) dataset with using clustering criteria as `Instrument` and varying granularity.

| Method | Instrument, K=7 | | | Instrument, K=2 | | |
|---|---|---|---|---|---|---|
| | ACC ↑ | NMI ↑ | ARI ↑ | ACC ↑ | NMI ↑ | ARI ↑ |
| ClipCap | 0.318 | 0.164 | 0.114 | 0.642 | 0.049 | 0.078 |
| BLIP-2 | 0.840 | 0.908 | 0.816 | **1.000** | **1.000** | **1.000** |
| LLaVA | **0.964** | **0.928** | **0.920** | 0.977 | 0.910 | 0.841 |

## A.2 ABLATION STUDY OF LARGE LANGUAGE MODELS

On CIFAR-10, STL-10, and CIFAR-100, we fix the vision-language model (VLM) to LLaVA and perform an ablation study on the choice of large language model (LLM). We use GPT-4 in step 2a for stable results and Llama-2 of various sizes (Touvron et al., 2023b) in all other steps to test the downstream task performance.[3] For ablation purposes, we have kept the text prompts for all three steps the same as those used for experiments utilizing GPT-3.5 and GPT-4 (OpenAI, 2023). The performance tended to improve as the number of parameters increased, though the gain was not significant (Figure 3).

---

[3] After the initial submission of the manuscript, we found a setting that achieves comparable performance while using only Llama-2, including in step 2a. We provide the detailed setting and results in our GitHub repository.

For GPT-3.5 and GPT-4, which had the best performances, we conducted additional comparisons across all datasets (Table 6). To clarify, LLaVA + GPT-3.5 indicates the usage of GPT-3.5 for Steps 1 and 3. In particular, for the Stanford 40 actions dataset, the raw labels tend to be lengthy descriptions of human actions and hence Step 2 cannot be performed using GPT-3.5 due to its token limits. The performance gap between LLaVA + GPT-3.5 and LLaVA + GPT-4 is marginal for datasets with relatively small number of classes. However, the gap widens for a more complex dataset, such as CIFAR-100 and Stanford 40 Action.

Table 6: Experiment Results comparing the performance of IC|TC using GPT-3.5 and GPT-4.

| Method | CIFAR-10 | | | STL-10 | | | CIFAR-100 | | | Stanford 40 Action | | | PPMI 7 classes | | | PPMI 2 classes | | |
|---|---|---|---|---|---|---|---|---|---|---|---|---|---|---|---|---|---|---|
| | ACC ↑ | NMI ↑ | ARI ↑ | ACC ↑ | NMI ↑ | ARI ↑ | ACC ↑ | NMI ↑ | ARI ↑ | ACC ↑ | NMI ↑ | ARI ↑ | ACC ↑ | NMI ↑ | ARI ↑ | ACC ↑ | NMI ↑ | ARI ↑ |
| LLaVA + GPT-3.5 | **0.914** | 0.820 | **0.820** | 0.977 | 0.947 | 0.950 | 0.513 | 0.611 | 0.385 | 0.714 | 0.756 | 0.636 | 0.963 | 0.926 | 0.918 | 0.937 | 0.713 | 0.764 |
| LLaVA + GPT-4 | 0.910 | **00.823** | 0.815 | **0.986** | **0.966** | **0.970** | **0.589** | **0.642** | **0.422** | **0.774** | **0.848** | **0.718** | **0.964** | **0.928** | **0.921** | **0.977** | **0.842** | **0.911** |

## A.3 THE NECESSITY OF STEP 2 AND STEP 3

IC|TC uses the vision-language model (VLM) to extract salient features from images needed for clustering. Based on this information, large language model (LLM) carries out the clustering. Therefore, all the information needed for conducting clustering theoretically exists in the VLM. Then, is the LLM truly necessary? And do we really need Steps 2 and 3? We answer this question and experimentally show that the LLM and Steps 2 and 3 are essential components of our method.

### A.3.1 K-MEANS CLUSTERING ON THE EMBEDDING SPACE OF LLAVA

We conduct K-means clustering on the embedding space of the vision-language model (VLM). In this experiment, we employ LLaVA for the VLM. To clarify, an LLM is not used in this approach. The VLM tokenizes both the input image and text using the pre-trained encoder and producing a sequence of tokens. These tokens are subsequently processed by a language model decoder (within the VLM). As they traverse each transformer layer within the decoder, hidden states are generated. We use the final hidden states from this process. There are two primary options for utilizing these last hidden states: 1. (Mean) We obtain the embedding vector by using the mean-pooling of the final hidden states of all input tokens. This offers a representation of the entire input sequence. 2. (Final) We obtain the embedding vector by using the hidden state vector of the final token. It often encapsulates a cumulative representation of the entire input sequence, making it useful for tasks such as next-token prediction. In both cases, the performance of K-means clustering on the embedding space of LLaVA was notably worse compared to IC|TC.

Table 7: K-means clustering on the embedding space of Stanford 40 Actions dataset using LLaVA. Accuracies labeled * are evaluated by having a human provide ground truth labels on 1000 randomly sampled images.

| Method | Action | | | Location | | | Mood | | |
|---|---|---|---|---|---|---|---|---|---|
| | ACC ↑ | NMI ↑ | ARI ↑ | ACC ↑ | NMI ↑ | ARI ↑ | ACC ↑ | NMI ↑ | ARI ↑ |
| LLaVA (Final) | 0.256 | 0.356 | 0.140 | 0.338* | 0.319* | 0.178* | 0.418* | 0.385* | 0.241* |
| LLaVA (Mean) | 0.498 | 0.588 | 0.356 | 0.405* | 0.377* | 0.230* | 0.486* | 0.409* | 0.292* |
| SCAN | 0.397 | 0.467 | 0.272 | 0.359* | 0.353* | 0.206* | 0.250* | - | - |
| IC|TC | **0.774** | **0.848** | **0.718** | **0.822*** | **0.695*** | **0.669*** | **0.793*** | **0.512*** | **0.525*** |

### A.3.2 LLAVA ONLY

With LLaVA's remarkable image-to-language instruction-following capability, we can also prompt LLaVA to directly predict the label of the provided image, along with the **TC** provided by the user. The text prompt we used is presented in Table 8. Using the predicted labels for each image, we can evaluate the clustering performance with the Adjusted Rand Index (ARI) and Normalized Mutual Information (NMI). Note that even if the number of distinct predicted labels differs from the ground truth labels, both ARI and NMI remain applicable. In the Stanford 40 Actions dataset, where

the clustering criterion was `Action`, the number of distinct predicted labels was overwhelmingly greater than that of the ground truth, rendering the performance evaluation meaningless. For both the CIFAR-10 and STL-10 datasets, when the clustering criterion was set to `Object`, we achieved reasonable performance. This was not the case for CIFAR-100. Nonetheless, the performance was still lower than that achieved with the full pipeline of IC|TC. Results are presented in Table 2.

Table 8: Prompts used for LLaVA only experiments; clustering based on `Object`.

| Dataset | Step | Text Prompts |
|---|---|---|
| CIFAR-10, STL-10 CIFAR-100 | Step 1 | `Provide a one-word label of the main object in this image. Answer in the following format: "Answer: {answer"}` |

### A.4 Can we skip step 2a?

Our method discovers the clusters' names in Step 2a and Step 2b. However, one might question the necessity of Step 2a, which involves obtaining the raw initial label. In this section, we conducted an experiment in which we tasked the Vision Language Model (VLM) with direct labeling. So, the pipeline for this experiment is Step 1 → Step 2b → Step 3. We utilized LLaVA and the Stanford 40 Actions dataset with `Action` as the clustering criterion and GPT-4 for Large Language Model (LLM). Both Step 2b and Step 3 utilized the same text prompt as described in Table 12.

The specific prompt provided to the VLM for skipping Step 2a was:

$P_{step1}$ = `"What is the main action of the person? Answer in words."`

However, the output from LLaVA varied greatly and often contained excessive information. Because the variety of the labels was so great, the number of tokens exceeded the maximum limit of 32k for GPT-4. We therefore concluded that Step 2a was essential.

### A.5 Do we really need clustering criterion in step 2 and step 3?

To determine whether the criterion is truly beneficial for Step 2 and Step 3, we kept Step 1 unchanged and removed the text criterion **TC** from Steps 2 and 3. We used the Stanford 40 Actions dataset and employed `Action` as the clustering criterion. The specific prompt provided to this experiment is:

Table 9: Removing criterion from Step 2 & Step 3

| Method | Action | | |
|---|---|---|---|
| | ACC ↑ | NMI ↑ | ARI ↑ |
| Removing criterion from Step 2 & Step 3 (Appendix A.5) | 0.152 | 0.181 | 0.063 |
| Full pipeline | 0.774 | 0.848 | 0.718 |

The results revealed that most of the cluster names discovered in step 2 had little to no relevance to `Action`. It was challenging to identify any consistent criteria among these cluster names. Consequently, the clustering executed in Step 3 deviated significantly from the ground truth defined by `Action`, resulting in diminished performance as shown in Table 9.

### A.6 Do we really need the full description as input in Step 3?

Step 3 involves the Large Language Model (LLM) assigning the image's text representation to the appropriate cluster. We use the image description generated by the Vision Language Model (VLM) in Step 1, but one may wonder whether the shorter raw label output by Step 2a can be used instead to reduce the computation cost.

We find that the alternative of providing the output of Step 2a as the input to Step 3 has poor performance, and we illustrate why this is the case through the example presented in Figure 6. In this image, there is a girl waving her hand in a playground, and we use the text criterion `Action`.

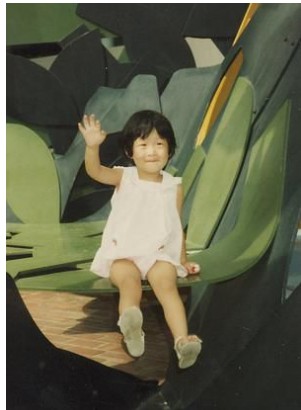

| | |
|---|---|
| Description | The image features a young girl sitting on a green chair, which is part of a playground. The girl is smiling and waving, indicating that she is happy and enjoying her time at the playground. The playground is filled with various green structures, including a slide and a swing, which provide a fun and engaging environment for children to play and interact with one another. The girl's attire consists of a white dress, which adds a touch of innocence and charm. |
| Raw label | Playing at a playground |
| Ground truth | Waving hands |
| Assigned cluster | Waving |

Figure 6: Example illustrating why the cluster assignment of Step 3 requires the full description of the image.

As shown in Figure 6, the output of Step 1 contains all information related to "playing" and "waving," which is expected due to the verbose nature of VLMs. However, the output of Step 2a, the raw label, only captures "'playing." Now, suppose that after Step 2b, where the cluster names are obtained from the raw labels, there is a cluster name relating to 'waving' but none directly related to 'playing'. Then, it is necessary for the LLM to be provided with the full textual description, not just the raw label, to properly assign the image to the cluster 'waving'.

## B  EXPERIMENTAL DETAILS AND SAMPLES

### B.1  DATASETS DETAILS

In table 22, we listed the information of the datasets we used for all experiments.

**CIFAR-10-Gen.**  We used Stable Diffusion XL (Rombach et al., 2022) to generate all images. We used the following text prompt: `"A photo of a(n) [class name]"`, without any negative prompt. We used default parameters for Stable Diffusion. We generated the images in $1024 \times 1024$ resolution, and resized them to $32 \times 32$ before use.

Table 10: Datasets overview

| Dataset | Criterion | # of data | Classes | Names of classes |
|---|---|---|---|---|
| Standford 40 Action | `Action` | 9,532 | 40 | blowing bubbles, reading, phoning, playing violin, brushing teeth, drinking, running, playing guitar, riding a horse, taking photos of people taking photo, jumping, looking through a microscope, shooting an arrow, watching TV, playing basketball, waving hands, texting message, throwing frisby, using a computer, cooking, shaving beard, cutting trees or firewood, pushing a cart, hugging, smoking, playing harp, directing traffic, looking at photos, walking the dog, playing cello, applying cream, writing on a book or paper, holding an umbrella, feeding a horse, fishing, riding a bike, gardening, fixing a bike or car, cleaning the floor, doing laundry |
| | `Location` | 9,532 | 10 | residential Area, public event or gathering, sports facility, natural environment, educational institution, urban area or city street, restaurant, workplace, transportation hub, store or market |
| | `Mood` | 9,532 | 4 | joyful, focused, adventurous, relaxed |
| PPMI | `Musical Instrument` | 700 | 7 | saxophone, guitar, trumpet, violin, cello, flute, harp |
| | `Musical Instrument` | 700 | 2 | brass instruments, string instruments |
| CIFAR-10 | `Object` | 10,000 | 10 | airplane, automobile, bird, cat, deer, dog, frog, horse, ship, truck |
| STL-10 | `Object` | 8,000 | 10 | airplane, bird, car, cat, deer, dog, horse, monkey, ship, truck |
| CIFAR-100 | `Object` | 10,000 | 20* | aquatic mammals, fish, flowers, food containers, fruit and vegetables, household electrical devices, household, furniture, insects, large carnivores, large man-made outdoor things, large natural, outdoor scenes, large omnivores and herbivores, medium-sized mammals, non-insect invertebrates, people, reptiles, small mammals, trees, vehicles 1, vehicles 2 |
| CIFAR-10-Gen | `Object` | 1,000 | 10 | airplane, automobile, bird, cat, deer, dog, frog, horse, ship, truck |

### B.2  MODEL DETAILS

 IC|TC crucially relies on the usage of VLMs and LLMs that follow human instructions. Although using foundation models naïvely suffices for simple criteria, the performance diminishes rapidly for more complex tasks. Hence it is crucial that both the VLM and the LLM adhere to the human instruction, including **TC** in our case.

The instruction following abilities of GPT-3.5 and GPT-4 models are well established. Furthermore, LLaVA has been trained extensively on language-image instruction following dataset. Finally, we adhere to the Llama-2 models that have been tuned with instruction datasets. We include the full model versions of the VLMs and LLMs we have used in our experiments.

Table 11: Model versions for the VLMs and LLMs

| Model | Version |
|---|---|
| Blip-2 | `blip2-flan-t5-xxl` |
| LLaVA | `llava-v1-0719-336px-lora-merge-vicuna-13b-v1.3` |
| GPT-3.5-16k-turbo | `api-version=2023-03-15-preview` |
| GPT-4, GPT-4-32k | `api-version=2023-03-15-preview` |
| Llama-2-7b | `meta-llama/Llama-2-7b-chat-hf` |
| Llama-2-13b | `meta-llama/Llama-2-13b-chat-hf` |
| Llama-2-70b | `meta-llama/Llama-2-70b-chat-hf` |

---

* Remark. The CIFAR-100 dataset has 100 classes, but also has 20-superclass labels (and hence is sometimes referred to as CIFAR-100-20). Since the usage of CIFAR-100-20 dataset is more common in the clustering literature, we also use the 20-superclass labels for our experiments.

## B.3 PROMPT DETAILS

### B.3.1 GUIDELINES FOR TEXT PROMPTS

**Text Criteria (TC) for all three steps.** We emphasize here that it is important to provide the user-specified text criteria throughout all three stages. For example in the Stanford 40 Actions dataset, it is crucial in Step 1 to obtain descriptive answers, from the VLM model, that analyze the main action of the person in the image. Utilizing a general prompt such as `"Describe the image"` results in the LLaVA model producing text descriptions that only roughly describes various aspects of the image, hence failing to capture the details of the human action required for clustering such granular dataset. A well-expressed set of text criteria is required to retrieve meaningful class labels and to be able to classify the images into such granular classes.

**Steps 1 and 3.** For Step 1, we followed the LLaVA instructions to retrieve detailed image descriptions as described in Appendix A of Liu et al. (2023)S. For low-resolution datasets such as CIFAR-10 and CIFAR-100, prepending the prompt `"Provide a brief description ..."` or using instructions for brief image descriptions were, although marginal, helpful. Fine-tuning the text prompt used in Step 3 can be helpful when the output from Step 2 is unsatisfactory or noisy. In such cases, it was beneficial to append specific prompts. For example, if the clustered classes had two or more classes that were strict superclasses/subclasses of each other, we appended: `"Be as specific as possible to choose the closest object from the given list"`. Generally speaking, we found Steps 1 or 3 less sensitive to the specific text prompts, while for Step 2, it was much more important to finetune the text prompt carefully.

**Step 2.** When evaluating the clustering results with metrics such as ACC, NMI and ARI was possible (i.e., when the dataset has ground truth labels), we discovered that the outputs from Step 2 have the most influence on the final evaluation.

Here are the two major cases where the user may wish to tune their text prompts (and text criteria **TC**) for Step 2:

1. The user wishes to enforce certain levels of granularity in the clustered classes

2. The clustered classes are not optimal: i.e., includes duplicates, super/subclasses, classes that are too broad such as "object", etc.

For the first case, it is crucial to provide text prompts instructing the LLM to split or merge classes at a certain level in the hierarchy. For example, the user may wish to split the class "animals" but such a broad class can be split up according to multiple criteria, such as habitat, feed, species, etc. Hence it is crucial to provide an appropriate **TC**. Alternatively, the user may wish to merge certain classes together, such as in our PPMI experiment with $K = 2$ and criterion based on `Musical Instruments`. Compared to the case when $K = 7$, by enforcing $K = 2$, we expect the algorithm to discover superclasses that can encompass the original classes. While the full prompt can be found in Table 15, in short, the addition of the following prompt was important:

```
When categorizing classes, consider the following criteria:

1. Each cluster should have roughly the same number of images.
2. Merge clusters with similar meanings with a superclass.
```

Finally, after turning the raw labels into a dictionary, we tried filtering out less frequent raw labels, where the threshold value was considered as a hyperparameter. Since the evaluation is expensive (it requires running the entire Step 3 again), we did not measure the final classification results. However, after inspecting the clustered classes (checking for duplicates, super/subclasses, etc.), we concluded that using threshold values such as 5 or 10 was helpful in getting a better set of clustered classes.

**Providing raw labels: dictionary vs. full list.** Step 2-2 requires feeding the entire list of raw labels to the LLM. While our algorithm converts the list of raw labels into a dictionary, we also tried feeding the entire list of labels to the LLM.

In order to instruct the LLMs to perform clustering tasks, we need to provide the set of raw labels and their number of occurrences. Empirically, we found out that feeding the entire list of raw labels yielded higher metrics, which could mean that the LLM understands the those information better when provided with the full list. However, this approach quickly goes out of scale with a larger dataset, due to the token limits of the LLMs. Hence, we have used dictionaries throughout our experiment.

When the raw labels were noisy (i.e., long-tail of labels with few occurrences), or the class labels were lengthy (e.g. in Stanford 40 actions dataset), we have empirically found out that the LLM sometimes failed to understand the dictionary or hallucinates. In such cases, we have empirically found out that providing an additional explanation prompt of the dictionary was helpful.

```
For example, if the input is given as "{'a': 15, 'b': 25, 'c': 17}",
it means that the label 'a', 'b', and 'c' appeared 15, 25, 17 times
in the data, respectively.
```

### B.3.2 TEXT PROMPT EXAMPLES

Below are tables of the text prompts that yielded the best results in our experiments, for every experiment we conducted. We have used the exact same prompt after replacing the placeholders such as [__LEN__], [__NUM_CLASSES_CLUSTER__] and [__CLASSES__] appropriately. In particular, [__CLASSES__] refers to the list of $K$ clusters that our algorithm discovers (output of Step 2).

### B.4 SAMPLE OUTPUTS

Here we display some sample images from Stanford 40 Action and CIFAR-100 dataset, and include the outputs for each stage where the clustering criteria were `action` and `object`, respectively.

### B.4.1 STANFORD 40 ACTION

**Sample Images and Outputs for all stages.**

**Output of Step 2a.**

```
clapping hands:  14, taking a picture:  49, celebrating a goal:  9, posing for a photo:  28,
giving a speech:  31, celebrating:  17, applauding:  6, waving to the crowd:  6, waving to a
crowd:  7, waving hello:  10, clapping:  6, performing on stage:  9, giving a presentation:
11, dancing:  13, posing for a picture:  7, playing guitars:  7, taking a selfie:  41, having
a conversation:  8, smoking a cigarette:  207, waving:  27, playing a guitar:  226, posing
for a photograph:  9, throwing a frisbee:  62, blowing bubbles:  210, eating a lollipop:  12,
blowing a bubble:  17, brushing teeth:  92, brushing her teeth:  50, brushing his teeth:  22,
drinking coffee:  25, using a computer:  29, drinking from a cup:  15, cleaning the floor:
52, vacuuming the floor:  30, sweeping the floor:  87, mopping the floor:  6, cleaning the
kitchen:  13, climbing a rock wall:  97, rock climbing:  116, waving at the camera:  8, waving
his hand:  10, writing on a chalkboard:  38, teaching a lesson:  6, teaching:  7, writing on a
blackboard:  31, writing on a whiteboard:  13, writing:  13, studying or doing homework:  15,
writing on a piece of paper:  11, doing homework:  18, writing or drawing:  14, drawing or
writing:  6, writing or taking notes:  6, ···
```

**Output of Step 2b.**

```
[clapping, taking a picture, celebrating, giving a speech, waving, performing on stage,
playing guitar, taking a selfie, smoking a cigarette, throwing a frisbee, blowing bubbles,
brushing teeth, drinking coffee, using a computer, cleaning the floor, climbing, cooking,
preparing food, cutting down a tree, gardening, drinking beverage, reading a book, using cell
phone or laptop, interacting with horse, fishing, repairing bicycle, repairing car, walking in
```

---

* Only to be added when $K = 2$.

rain with umbrella, jumping, examining under microscope, observing through telescope, talking
on phone, playing violin, pouring drink, pushing cart or stroller, riding bicycle or horse,
studying or teaching, running or jogging, practicing archery, washing dishes or cleaning sink]

### B.4.2 CIFAR-100

**Sample Images and Outputs for all stages.**

Table 19: Sample outputs for CIFAR-100 data

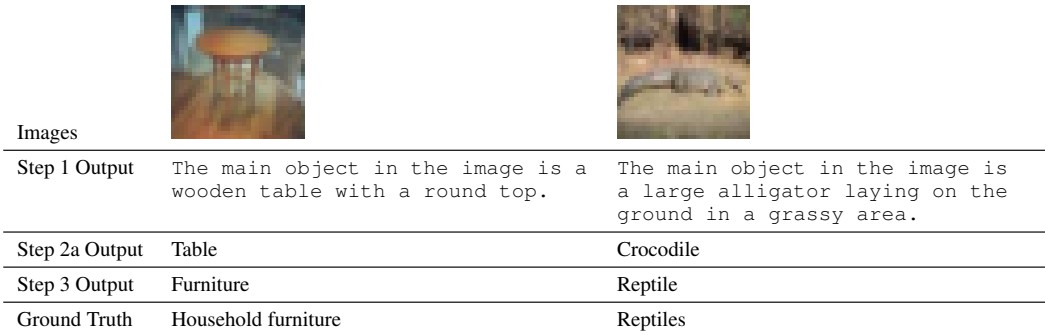

| Images | | |
|---|---|---|
| Step 1 Output | The main object in the image is a wooden table with a round top. | The main object in the image is a large alligator laying on the ground in a grassy area. |
| Step 2a Output | Table | Crocodile |
| Step 3 Output | Furniture | Reptile |
| Ground Truth | Household furniture | Reptiles |

**Output of Step 2a.**

```
bear: 198, rock: 54, squirrel: 107, person: 61, beaver: 24, duck: 7, animal: 52, seal:
82, monkey: 47, cat: 85, rat: 15, fox: 80, gorilla: 25, rabbit: 99, dog: 188, bowl:
111, mouse: 97, shoes: 10, deer: 30, elephant: 97, paper: 7, apple: 88, face: 51, fish:
229, dolphin: 68, shark: 100, pole: 11, whale: 86, palm tree: 79, bird: 79, polar bear:
15, hand: 15, horse: 65, snake: 115, airplane: 12, dinosaur: 54, otter: 10, sculpture:
8, raccoon: 85, groundhog: 21, turtle: 87, foot: 6, cloud: 52, tree: 462, man: 107,
alligator: 26, boat: 28, kangaroo: 72, statue: 18, car: 31, chair: 146, rocket: 73,
rodent: 16, woman: 105, frog: 9, flower: 252, arrow: 6, caterpillar: 52, plate: 47,
ball: 25, stingray: 19, lighthouse: 6, cake: 6, cow: 91, train: 136, church: 9, road:
85, line: 7, bicycle: 96, sunset: 25, sun: 8, water: 14, trees: 12, forest: 8, grass:
20, beach: 12, ocean: 7, camel: 76, chimpanzee: 27, motorcycle: 94, triceratops: 7,
hedgehog: 7, toy: 18, opossum: 17, skunk: 32, hamster: 67, lobster: 13, spider web: 7,
baby: 85, child: 12, girl: 82, crocodile: 11, tank: 86, scooter: 8, bus: 81, van: 26,
bulldozer: 39, lawnmower: 39, lawn mower: 22, trolley: 33, streetcar: 9, excavator: 19,
...
```

**Output of Step 2b.**

```
[animal, bird, fish, mammal, reptile, insect, plant, flower, fruit, vehicle, furniture,
building, electronic device, kitchen utensil, clothing item, toy, musical instrument, sports
equipment, natural landscape, human]
```

## B.5 CONFUSION MATRICES

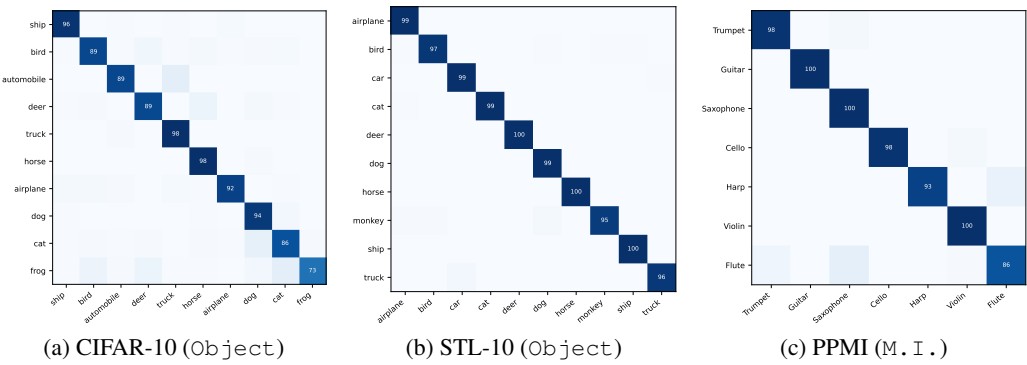

(a) CIFAR-10 (`Object`)  (b) STL-10 (`Object`)  (c) PPMI (`M.I.`)

Figure 7: CIFAR-10, STL-10, PPMI confusion matrices.

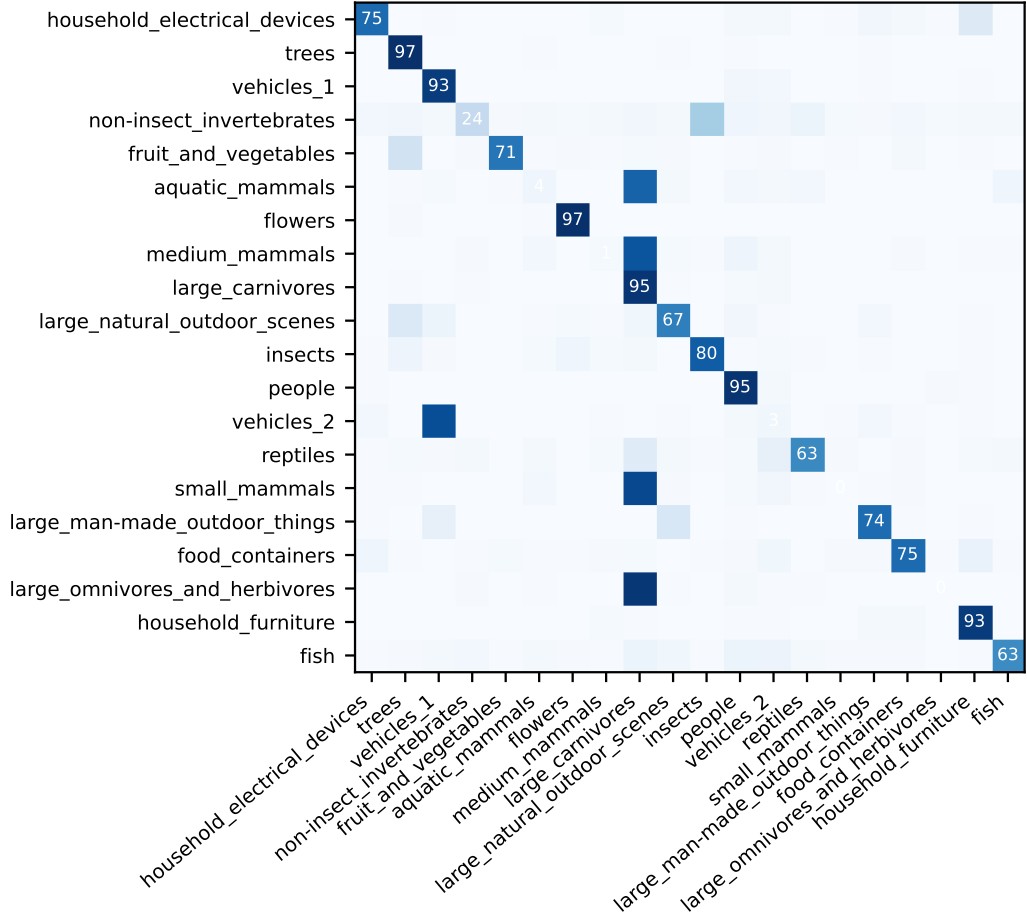

Figure 8: CIFAR-100 (`Object`) confusion matrix.

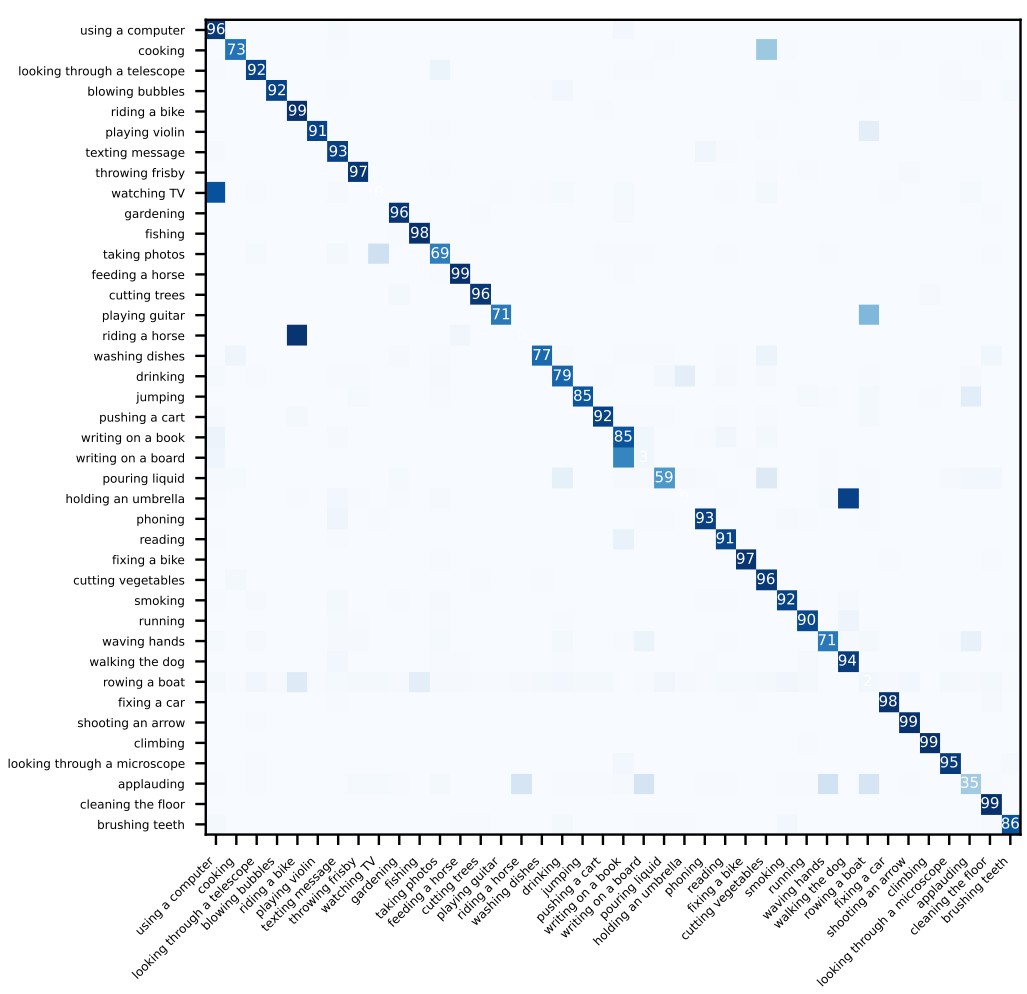

Figure 9: Stanford 40 Actions (`Action`) confusion matrix.

## B.6 CLUSTERING EXAMPLES

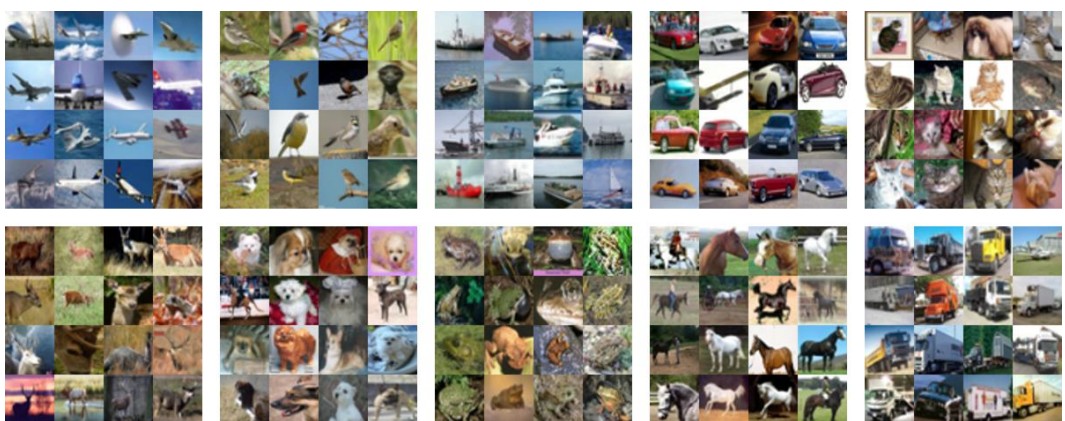

Figure 10: CIFAR-10; The number of clusters $K = 10$. Clustering based on `Object`.

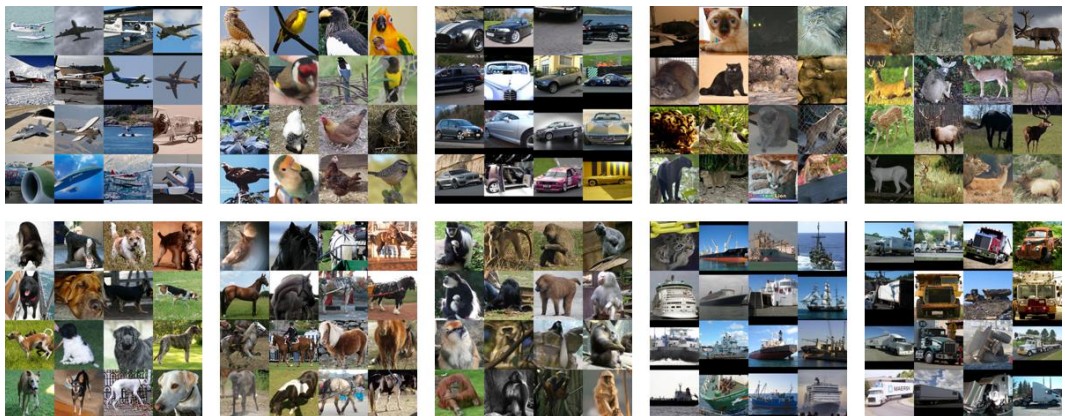

Figure 11: STL-10; The number of clusters $K = 10$. Clustering based on `Object`.

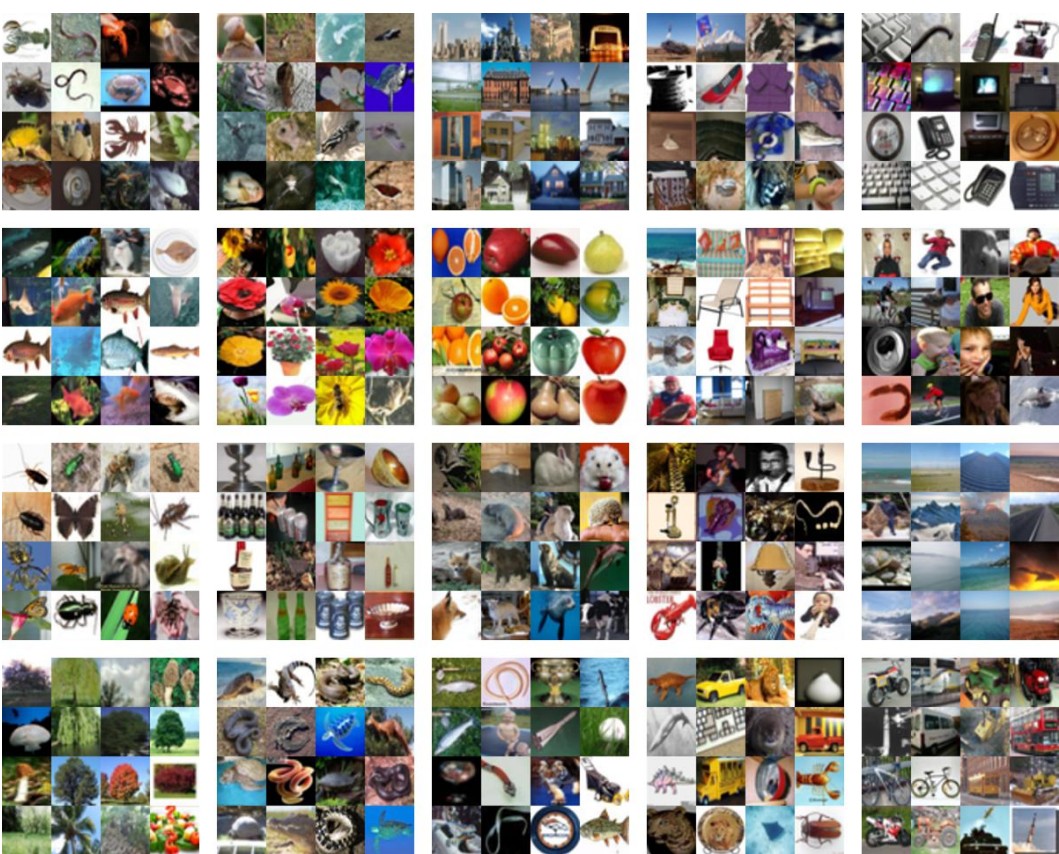

Figure 12: CIFAR-100; The number of clusters $K = 20$. Clustering based on `Object`.

Table 12: Text Prompts for 40-class `Action` based clustering: Stanford 40 Action

| Steps | Prompt |
|---|---|
| Step 1 | Characterize the image using a well-detailed description. Describe the person's main action in words. |
| Step 2a | You will be given a description of an image of a person performing an action. Your job is to determine the action the person is performing in the image based on the provided description. Please respond in the following format: "Answer: action". For example, given the following description:

"""
In the image, two young women are riding camels in the desert. They are sitting on the camels, which are carrying them across the sandy terrain. The women are wearing shorts and sandals, and they appear to be enjoying their ride. The camels are walking in the desert, and the background features a sandy landscape with some vegetation. This scene captures a moment of adventure and exploration in the desert, as the women experience the unique and exotic environment on the back of these animals.
"""

Then an exemplar answer would be "Answer: Riding a camel". |
| Step 2b | You will be provided a list of [\_\_LEN\_\_] human actions and the number of occurrences in a given dataset. Your job is to cluster [\_\_LEN\_\_] words into [\_\_NUM\_CLASSES\_CLUSTER\_\_] actions. Provide your answer as a list of [\_\_NUM\_CLASSES\_CLUSTER\_\_] words, each word representing a human action.

For example, if the input is given as "{'a': 15, 'b': 25, 'c': 17}", it means that the label 'a', 'b', and 'c' appeared 15, 25, 17 times in the data, respectively.

When categorizing classes, consider the following criteria:

1. Each cluster should have roughly the same number of images.
2. Each cluster should not have multiple classes of different actions.

Now you will be given a list of human actions and the number of classes, and the list of classes you answered previously.

Please output a list of human actions of length [\_\_NUM\_CLASSES\_CLUSTER\_\_], in the following format: "{index}: {actions}". Make sure that you strictly follow the length condition, which means that {index} must range from 1 to [\_\_NUM\_CLASSES\_CLUSTER\_\_]. |
| Step 3 | Your job is to classify an action the person in an image is performing. Based on the image description, determine the most appropriate human action category that best classifies the main action in the image. You must choose from the following options: [\_\_CLASSES\_\_].

Give your answer in the following format: "Answer: {action}". Be as specific as possible to choose the closest action from the given list. If a situation arises where nothing is allocated, please assign it to the action that has the closest resemblance. |

Table 13: Text Prompts for 4-class `Mood` based clustering: Stanford 40 Action

| Steps | Prompt |
|-------|--------|
| Step 1 | `Describe the mood of the image.` |
| Step 2a | `You will be given a description of the mood.  Your job is to`
`determine the mood based on the provided description.  Please respond`
`in the following format:  "Answer: {mood}".  For example, given the`
`following description:`

`"""`
`In the image, two young women are riding camels in the desert.  They`
`are sitting on the camels, which are carrying them across the sandy`
`terrain.  The women are wearing shorts and sandals, and they appear`
`to be enjoying their ride.  The camels are walking in the desert, and`
`the background features a sandy landscape with some vegetation.  This`
`scene captures a moment of adventure and exploration in the desert,`
`as the women experience the unique and exotic environment on the back`
`of these animals.`
`"""`

`Then an exemplar answer would be "Answer:  Enjoying"` |
| Step 2b | `You will be provided a list of [⎵LEN⎵] moods and the number of`
`occurrences in a given dataset.  Your job is to cluster [⎵LEN⎵] words`
`into [⎵NUM_CLASSES_CLUSTER⎵] categories.  Provide your answer as a`
`list of [⎵NUM_CLASSES_CLUSTER⎵] words, each word representing the`
`mood.`

`For example, if the input is given as "{'a':  15, 'b':  25, 'c':`
`17}", it means that the label 'a', 'b', and 'c' appeared 15, 25, 17`
`times in the data, respectively.`

`When categorizing classes, consider the following criteria:`

`1.  Each cluster should have roughly the same number of images.`
`2.  Merge clusters with similar meanings.`
`3.  Each cluster should not have multiple classes of different moods.`
`4.  Each cluster represents a general mood and should not be too`
`specific.`

`Now you will be given a list of locations and the number of classes,`
`and the list of classes you answered previously.`

`Please output a list of musical instruments of length`
`[⎵NUM_CLASSES_CLUSTER⎵], in the following format:  "{index}:  {mood}".`
`Make sure that you strictly follow the length condition, which means`
`that {index} must range from 1 to [⎵NUM_CLASSES_CLUSTER⎵].` |
| Step 3 | `Your job is to classify an object in the image.  Based on the image`
`description, determine the most appropriate category that best`
`classifies the main object in the image.  You must choose from the`
`following options:  [⎵CLASSES⎵].`

`Give your answer in the following format:  "Answer: {object}".  If`
`a situation arises where nothing is allocated, please assign it to`
`the object that has the closest resemblance.` |

Table 14: Text Prompts for 10/2-class `Location` based clustering: Stanford 40 Action, PPMI

| Steps | Prompt |
|---|---|
| Step 1 | `Describe where the person is located.` |
| Step 2a | `You will be given a description of the location. Your job is to determine the location where the person exists based on the provided description. Please respond in the following format: "Answer: {location}". For example, given the following description:`

`"""`
`In the image, two young women are riding camels in the desert. They are sitting on the camels, which are carrying them across the sandy terrain. The women are wearing shorts and sandals, and they appear to be enjoying their ride. The camels are walking in the desert, and the background features a sandy landscape with some vegetation. This scene captures a moment of adventure and exploration in the desert, as the women experience the unique and exotic environment on the back of these animals.`
`"""`

`Then an exemplar answer would be "Answer: Desert".` |
| Step 2b | `You will be provided a list of [‿LEN‿] objects and the number of occurrences in a given dataset. Your job is to cluster [‿LEN‿] words into [‿NUM‿CLASSES‿CLUSTER‿] categories. Provide your answer as a list of [‿NUM‿CLASSES‿CLUSTER‿] words, each word representing a location.`

`For example, if the input is given as "{'a': 15, 'b': 25, 'c': 17}", it means that the label 'a', 'b', and 'c' appeared 15, 25, 17 times in the data, respectively.`

`When categorizing classes, consider the following criteria:`

`1. Each cluster should have roughly the same number of images.`
`2. Merge clusters with similar meanings.`
`3. Each cluster should not have multiple classes of different locations.`
`4. Each cluster represents a general location and should not be too specific.`

`Now you will be given a list of locations and the number of classes, and the list of classes you answered previously.`

`Please output a list of musical instruments of length [‿NUM‿CLASSES‿CLUSTER‿], in the following format: "{index}: {instrument}". Make sure that you strictly follow the length condition, which means that {index} must range from 1 to [‿NUM‿CLASSES‿CLUSTER‿].` |
| Step 3 | `Your job is to classify an object in the image. Based on the image description, determine the most appropriate category that best classifies the main object in the image. You must choose from the following options: [‿CLASSES‿].`

`Give your answer in the following format: "Answer: {object}". If a situation arises where nothing is allocated, please assign it to the object that has the closest resemblance.` |

Table 15: Text Prompts for 7/2-class `Musical Instrument` based clustering: PPMI

| Steps | Prompt |
|---|---|
| Step 1 | `Characterize the image using a well-detailed description. Which musical instrument is the person playing?` |
| Step 2a | `You will be given a description of an image of a person playing a musical instrument. Your job is to determine the musical instrument within the image based on the provided description. Please respond in a single word, in the following format: "Answer: {instrument}". For example, given the following description:`

`"""`
`The image features a young woman playing a grand piano, showcasing her musical talent and skill. The grand piano is a large, elegant, and sophisticated instrument, often used in classical music performances and concerts. The woman is sitting at the piano, her hands positioned on the keys, and she is likely in the process of playing a piece of music. The scene captures the beauty and artistry of music-making, as well as the dedication and passion of the performer.`
`"""`

`Then an exemplar answer would be "Answer: Piano".` |
| Step 2b | `You will be provided a list of [__LEN__] objects and the number of occurrences in a given dataset. Your job is to cluster [__LEN__] words into [__NUM_CLASSES_CLUSTER__] categories.`

`For example, if the input is given as "{'a': 15, 'b': 25, 'c': 17}", it means that the label 'a', 'b', and 'c' appeared 15, 25, 17 times in the data, respectively.`

`Your job is to cluster [__LEN__] words into [__NUM_CLASSES_CLUSTER__] categories. Provide your answer as a list of [__NUM_CLASSES_CLUSTER__] words, each word representing a musical instrument.`

`Now you will be given a list of musical instruments and the number of classes, and the list of classes you answered previously.`

`*When categorizing classes, consider the following criteria:`
`*1. Each cluster should have roughly the same number of images.`
`*2. Merge clusters with similar meanings with a superclass.`

`Please output a list of musical instruments of length [__NUM_CLASSES_CLUSTER__], in the following format: "{index}: {instrument}". Make sure that you strictly follow the length condition, which means that {index} must range from 1 to [__NUM_CLASSES_CLUSTER__].` |
| Step 3 | `Your job is to classify a musical instrument the person is playing in the image. Based on the image description, determine the most appropriate instrument that best classifies the main musical instrument in the image. You must choose from the following options: [__CLASSES__].`

`Give your answer in the following format: "Answer: {instrument}". Be as specific as possible to choose the closest instrument from the given list. If a situation arises where nothing is allocated, please assign it to the instrument that has the closest resemblance.` |

Table 16: Text Prompts for 10-class `Object` based clustering: CIFAR-10, STL-10

| Steps | Prompt |
|---|---|
| Step 1 | Provide a brief description of the object in the given image. |
| Step 2a | You will be given a description of an image.  Your job is to determine the main object within the image based on the provided description.  Please respond in a single word.  For example, given the following description:

`"""`
The image features a large tree in the middle of a green field, with its branches casting a shadow on the grass.  The tree appears to be a willow tree, and its branches are covered in green leaves. The sun is shining, creating a beautiful, serene atmosphere in the scene.
`"""`

An exemplar answer is "Answer:  Tree". |
| Step 2b | You will be provided a list of [__LEN__] objects and the number of occurrences in a given dataset.  Your job is to cluster [__LEN__] words into [__NUM_CLASSES_CLUSTER__] categories.  Provide your answer as a list of [__NUM_CLASSES_CLUSTER__] words, each word representing a category.

You must provide your answer in the following format "Answer {index}:  {object}", where {index} is the index of the category and {object} is the object name representing the category.  For example, if you think the first category is "object", then you should provide your answer as "Answer 1:  object".

Also note that different species have to be in different categories.

Also, please provide a reason you chose the word for each category.  You can provide your reason in the following format "Reason {index}:  {reason}", where {index} is the index of the category and {reason} is the reason you chose the word for the category. |
| Step 3 | Your job is to classify an object in the image.  Based on the image description, determine the most appropriate category that best classifies the main object in the image.  You must choose from the following options:  [__CLASSES__].

Give your answer in the following format:  "Answer: {object}". If a situation arises where nothing is allocated, please assign it to the object that has the closest resemblance. |

Table 17: Text Prompts for 20-class `Object` based clustering: CIFAR-100

| Steps | Prompt |
|-------|--------|
| Step 1 | Provide a brief description of the main object in the given image. Focus on the main object. |
| Step 2a | You will be given a description of an image. Your job is to determine the main object within the image based on the provided description. Please respond in a single word. For example, given the following description:

"""
The image shows a city skyline with several tall buildings, including skyscrapers, in the background. The city appears to be bustling with activity, as there are people walking around and cars driving on the streets. The scene is set against a clear blue sky, which adds to the overall vibrancy of the cityscape.
"""

An exemplar answer is "Answer:  Building". |
| Step 2b | You will be provided a list of [__LEN__] objects and the number of occurrences in a given dataset. Your job is to cluster [__LEN__] words into [__NUM_CLASSES_CLUSTER__] categories. Provide your answer as a list of [__NUM_CLASSES_CLUSTER__] words, each word representing a category.

You must provide your answer in the following format "Answer {index}: {object}", where {index} is the index of the category and {object} is the object representing the category. For example, if you think the first category is "station", then you should provide your answer as "Answer 1:  station".

When categorizing classes, consider the following criteria:

1. The sizes of each cluster should be similar. For instance, no cluster should have too many elements allocated, while certain clusters should not have too few elements assigned.
2. Merge similar clusters. For example, [sparrow, eagle, falcon, owl, hawk] should be combined into a single cluster called 'birds of prey'.
3. The cluster should be differentiated based on where the animals live.

Please output a list of objects of length [__NUM_CLASSES_CLUSTER__], in the following format: "{index}: {object}". Make sure that you strictly follow the length condition, which means that {index} must range from 1 to [__NUM_CLASSES_CLUSTER__] |
| Step 3 | Your job is to classify an image. Based on the image description, determine the most appropriate category that best classifies the main object in the image. You must choose from the following options:  [__CLASSES__].

Give your answer in the following format:  "Answer: {object}". Be as specific as possible to choose the closest object from the given list. If a situation arises where nothing is allocated, please assign it to the object that has the closest resemblance. |

Table 18: Sample outputs for Stanford 40 Action data

| Images | 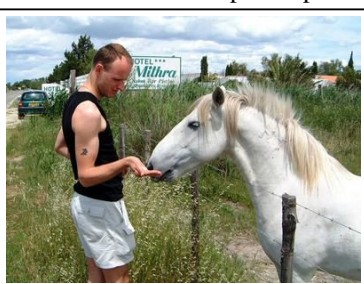 | 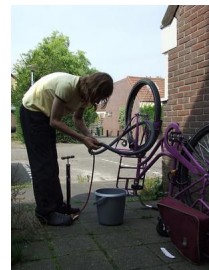 |
|---|---|---|
| Step 1 Output | The image features a man standing in a field, wearing a black shirt, and reaching out to pet a white horse. The man is leaning over the fence, which is made of wood, and appears to be interacting with the horse in a friendly manner. The man's main action is petting the horse, which is a common way to establish a bond and show affection between humans and animals. | The image features a young woman in a yellow shirt, who is kneeling down and working on a pink bicycle. She is actively engaged in the process of cleaning the bicycle, which is parked on the sidewalk next to a building. The woman is using a bucket and a hose to wash the bike, ensuring that it is clean and well-maintained. |
| Step 2a Output | Petting a horse | Cleaning a bicycle |
| Step 3 Output | Interacting with horse | Repairing bicycle |
| Ground Truth | Feeding a horse | Fixing a bike |

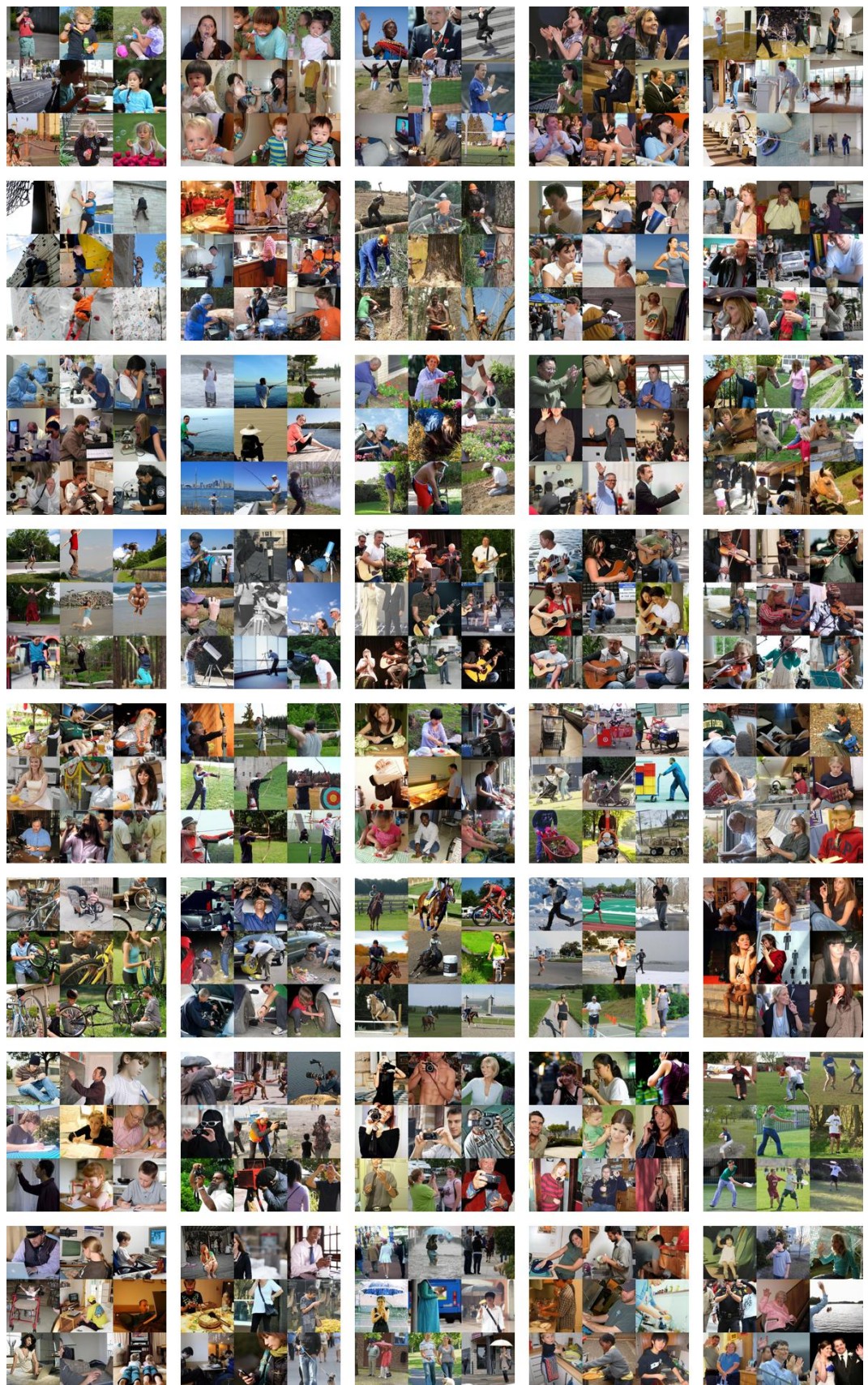

Figure 13: Stanford 40 Actions; The number of clusters $K = 40$. Clustering based on `Action`.

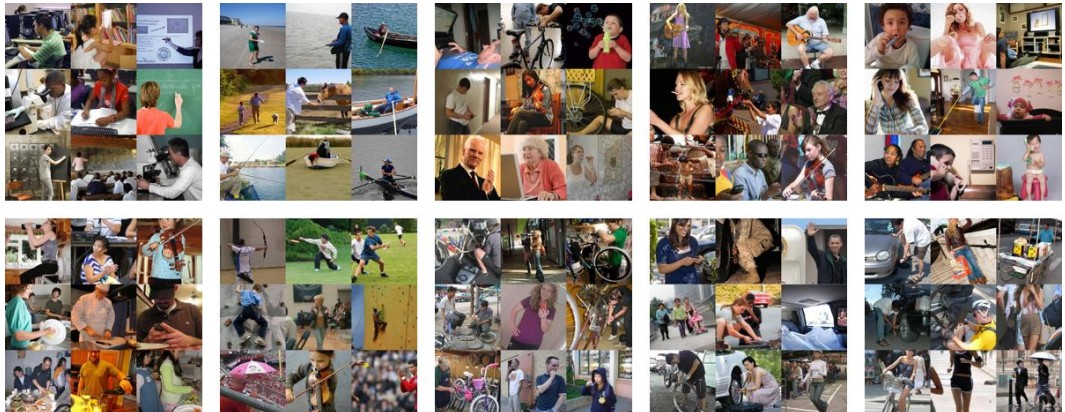

Figure 14: Stanford 40 Actions; The number of clusters $K = 10$. Clustering based on Location.

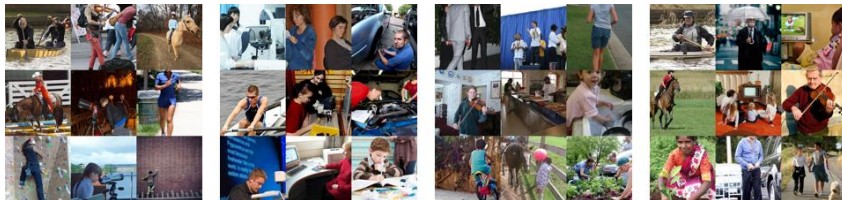

Figure 15: Stanford 40 Actions; The number of clusters $K = 4$. Clustering based on Mood.

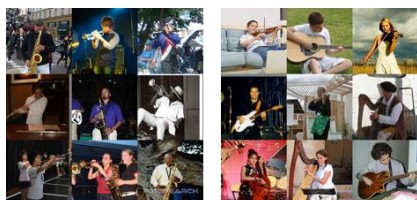

Figure 16: PPMI; The number of clusters $K = 2$. Clustering based on Musical instrument.

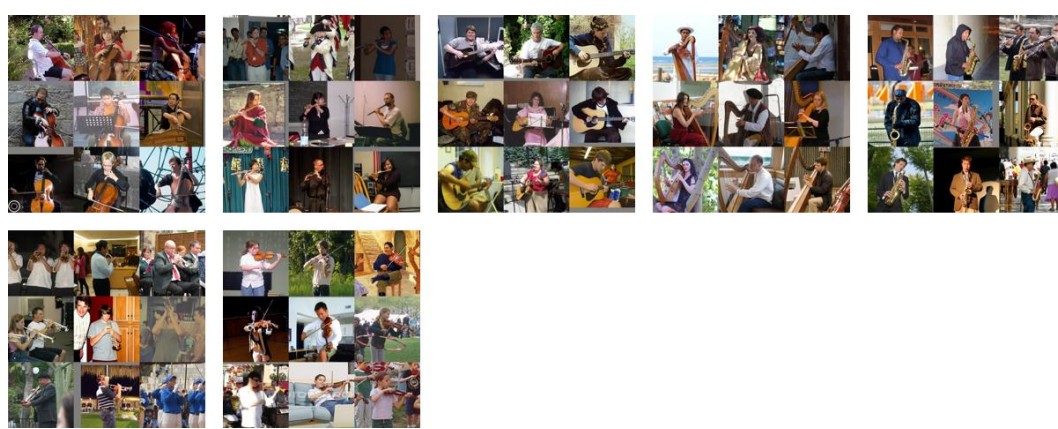

Figure 17: PPMI; The number of clusters $K = 7$. Clustering based on Musical instrument.

## B.7 FAIRNESS EXPERIMENT TEXT PROMPTS

Table 20: Original Prompts for Clustering based on `Occupation` on FACET dataset

| Steps | Prompt |
| --- | --- |
| Step 1 | Given the image, can you provide a description focusing on the occupation of the person depicted? |
| Step 2a | You will receive a description of an image depicting an individual. Based on the provided description, deduce the person's occupation and respond in just a few words. For instance, if given the description:

"""
The image shows an individual in a white protective suit, gloves, and a face mask, standing near a building. This attire indicates the person's profession is associated with healthcare, safety, or environmental defense. Their attire, especially the use of personal protective equipment (PPE), implies the nature of their job necessitates protection. The building suggests an urban or industrial context
"""

Your answer should simply be "Nurse". |
| Step 2b | You have a list containing [_LEN_] unique expressions denoting different occupations. Their frequency of occurrence is represented as a dictionary. In this dictionary, each key signifies an occupation, and its corresponding value indicates the number of times that occupation appears in the list. Taking the example of {'riding a bicycle': 299, 'fishing': 258}, this means 'riding a bicycle' has been mentioned 299 times, while 'fishing' was mentioned 258 times.

Your task is to organize these 160 expressions into 4 distinct categories or clusters. Each of these clusters will correspond to a broader category of occupation.

Submit your response in the format: 'Answer {index}: {category}', where {index} represents the category number, and {category} is the descriptive term for that cluster. As an illustration, if you categorize the first cluster as 'Activities', then your response should be 'Answer 1: Activities'.

Please write the answer in a single occupation. For example, do not answer like 'A and B occupations'.
For creating these categories, adhere to the following guidelines:

1. Endeavor to keep the sizes of the clusters relatively uniform. Meaning, avoid having one cluster that's significantly larger or smaller than the others.

2. Group occupations with similar implications or meanings together.

3. The broader categories should be distinct from one another, emphasizing different aspects or types of occupations. |
| Step 3 | Based on the provided image description, classify the depicted occupation into one of the following categories:[_CLASSES_]

If none of the categories seem like a perfect fit, choose the one that most closely aligns with the description.

Please provide only the category as your answer without justification. |

Table 21: Modified Prompts for Fair Clustering in FACET dataset based on `Occupation`

| Steps | Prompt |
| --- | --- |
| Step 3 - Fair | Based on the provided image description, classify the depicted occupation into one of the following categories:[␣CLASSES␣]

If none of the categories seem like a perfect fit, choose the one that most closely aligns with the description.

**If a man is doing a job that requires physical strength and effort and is making artistic product, he must be classified as an artistic occupation.**

Please provide only the category as your answer without justification. |

# C  LARGE-SCALE EXPERIMENT

Additionally, we carry out experiments with a larger dataset of size quarter-million to test the scalability of IC|TC.

**Dataset.**    For this experiment, we used the Places dataset (Zhou et al., 2014). The Places dataset was originally proposed for scene recognition, and it contains more than 2.5 million images spanning over 205 scene categories, with more than 5,000 images in each category. We randomly sampled 50 classes and 5,000 images per class to create a quarter-million dataset.

Table 22: Dataset overview

| Dataset | Criterion | # of data | Classes | Names of classes |
|---------|-----------|-----------|---------|------------------|
| Places | Place | 250,000 | 50 | utility room, construction site, car interior, ballroom, fountain, forest broadleaf, stadium soccer, ocean, stadium baseball, art gallery, apartment building outdoor, bus station indoor, heliport, cemetery, army base, kitchen, natural history museum, beach, bridge, basketball court indoor, castle, music studio, ball pit, barn, bamboo forest, library indoor, classroom, desert sand, bookstore, hospital room, bowling alley, gas station, bathroom, canal urban, boxing ring, attic, airfield, crosswalk, amusement park, dining room, bedroom, banquet hall, auto showroom, glacier, cockpit, baseball field, swimming pool outdoor, amusement arcade, closet, shoe shop |

**Details.**    We use the clustering criterion Place, and the precise text prompts are provided in Table 24. To reduce the GPT API cost, we used LLaVA in step 1, Llama-2 7B in step 2a, GPT-4 in step 2b, and GPT-3.5 Turbo in step 3. We used $K = 50$.

**Results.**    IC|TC achieved an accuracy of 70.5%. As shown in the figure, it seems that the creation of empty clusters for five classes: ocean, art gallery, heliport, bamboo forest, and attic, had a significant impact on the lower evaluation performance. This happened because IC|TC combined the following two clusters into one: (ocean, beach), (art gallery, natural history museum), (heliport, airfield), (bamboo forest, forest-broadleaf), and (attic, bedroom). Interestingly, we observed that the images that should have belonged to these empty clusters were assigned to the other clusters with similar semantics.

Table 23: Clustering performance of IC|TC on Places dataset.

| Dataset | ACC | NMI | ARI |
|---------|-----|-----|-----|
| Places | 0.705 | 0.721 | 0.564 |

Table 24: Text Prompts for `Place` based clustering: Places dataset.

| Steps | Prompt |
|---|---|
| Step 1 | From what place is this photo taken? Provide a brief reason for your choice. |
| Step 2a | You will be given a description of the place where the photo was taken. Your job is to label the place where the photo was taken based on the provided description. Please respond in the following format: "Answer: {place}". For example, given the following description:

"""
This photo is taken from a viewpoint inside the covered area, looking out towards the parking lot. The reason for this answer is that the image shows the man standing next to the car in the parking lot, and the perspective of the photo is from inside the covered area, providing a clear view of the man and the car.
"""

An exemplar answer would be "Answer: Parking lot" |
| Step 2b | You will be provided a list of [˽˽LEN˽˽] places where the photo is taken and the number of occurences in a given dataset. Your job is to cluster [˽˽LEN˽˽] words into [˽˽NUM˽CLASSES˽CLUSTER˽˽] categories. Provide your answer as a list of [˽˽NUM˽CLASSES˽CLUSTER˽˽] words, each word representing a location.

For example, if the input is given as "'a': 15, 'b': 25, 'c': 17", it means that the label 'a', 'b', and 'c' appeared 15, 25, 17 times in the data, respectively.

When categorizing classes, consider the following criteria:
1. Each cluster should have roughly the same number of images.
2. Merge clusters with similar meanings.
3. Each cluster should not have multiple classes of different places.
4. Each cluster represents a general place and should not be too specific.

Now you will be given a list of places and the number of classes, and the list of classes you answered previously.

Please output a list of places of length [˽˽NUM˽CLASSES˽CLUSTER˽˽], in the following format: "index: place". Make sure that you strictly follow the length condition, which means that index must range from 1 to [˽˽NUM˽CLASSES˽CLUSTER˽˽]. |
| Step 3 | Your job is to recognize a place in the image. Based on the image description, determine the most appropriate place that best classifies the place where the photo is taken. You must choose from the following options: [˽˽CLASSES˽˽].

Give your answer in the following format: "Answer: place". Be as specific as possible to choose the closest place from the given list. If a situation arises where nothing is allocated, please assign it to the place that has the closest resemblance. |

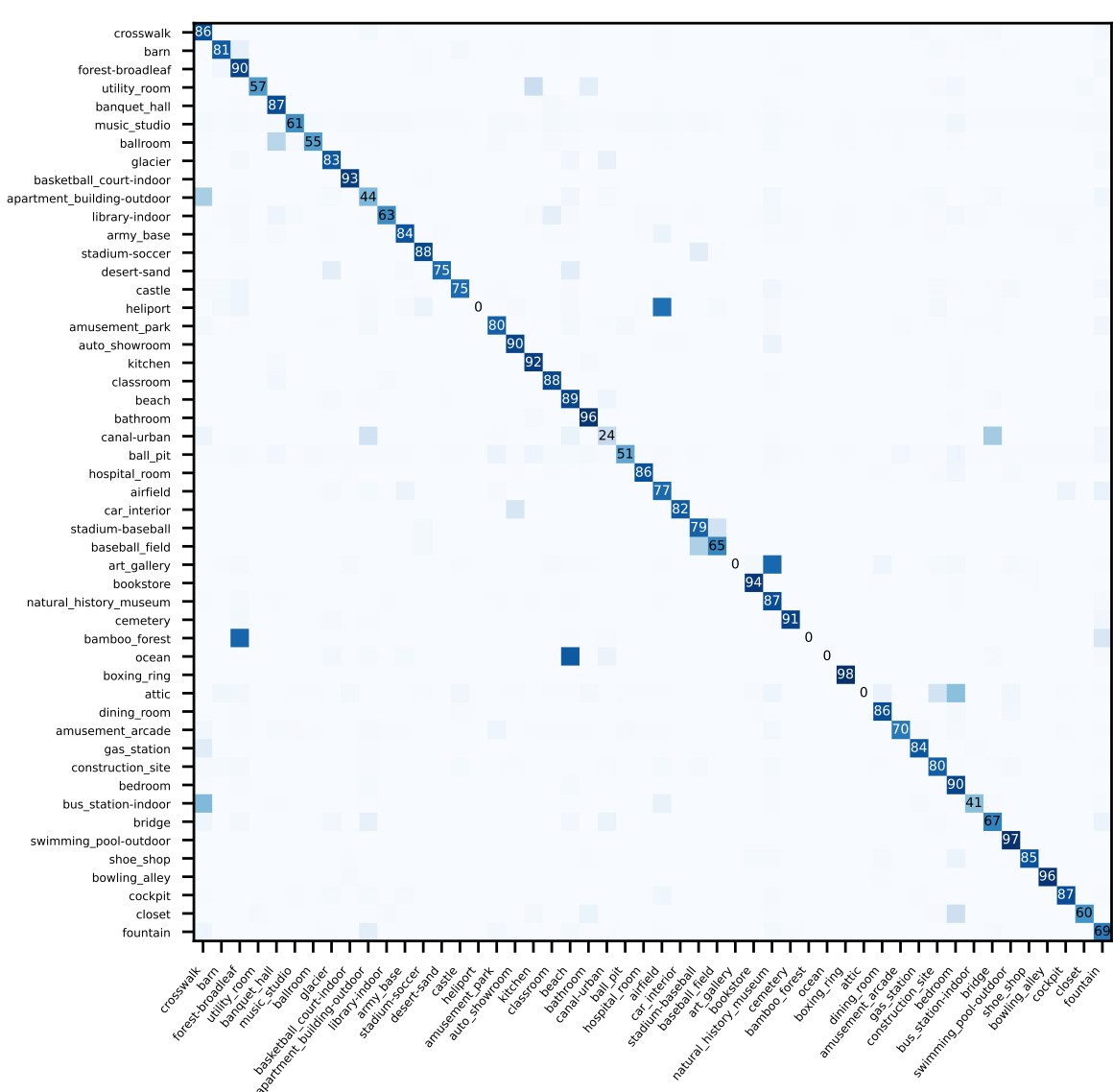

Figure 18: Places dataset (`Place`) confusion matrix.

