# OpenReview forum: "Image Clustering Conditioned on Text Criteria"
_ICLR.cc/2024/Conference — ICLR 2024 poster_

### Official Review · Reviewer_heiR · 2023-10-16

**Soundness:** 3 good
**Presentation:** 4 excellent
**Contribution:** 3 good
**Rating:** 8
**Confidence:** 3

**Summary:**

The authors proposed a new image clustering paradigm IC|TC, which performing image clustering based on user-specified criteria in the form of text by leveraging modern Vision-Language Models and Large Language Models.  IC|TC can effectively cluster images with various criteria, such as human action, physical location, or the person’s mood, significantly outperforming baselines.

**Strengths:**

1. The paper is well organized and the presentation is pretty good.
2. The paradigm of IC|TC is new
3. The experimental part is detailed and the clustering results are amazing

**Weaknesses:**

1. Since the model size of LLaMA-2 and GPT-4 are too big, the author can give more testing results on smaller models.
2. Some baseline models are missing, such as GCC[1] and TCC[2]

[1]Zhong H, Wu J, Chen C, et al. Graph contrastive clustering. Proceedings of the IEEE/CVF international conference on computer vision. 2021: 9224-9233.

[2]Shen Y, Shen Z, Wang M, et al. You never cluster alone[J]. Advances in Neural Information Processing Systems, 2021, 34: 27734-27746.

**Questions:**

1. Since the model size of LLaMA-2 and GPT-4 are too big, the author can give more testing results on smaller models.
2. Some baseline models are missing, such as GCC[1] and TCC[2]

[1]Zhong H, Wu J, Chen C, et al. Graph contrastive clustering. Proceedings of the IEEE/CVF international conference on computer vision. 2021: 9224-9233.

[2]Shen Y, Shen Z, Wang M, et al. You never cluster alone[J]. Advances in Neural Information Processing Systems, 2021, 34: 27734-27746.

---

> ### Author Response · Authors · 2023-11-21
> **Individual response to Reviewer heiR**
>
> We are happy to hear that the reviewer appreciates our experimental results and our paradigm's novelty.
>
>
> $\\phantom{X}$
>
> **Response to "Since the model size of LLaMA-2 and GPT-4 are too big, the author can give more testing results on smaller models."**
>
> We conducted an ablation study on object clustering using Llama 2 7B, 13B, 70B, GPT-3.5, and GPT-4 (Figure 3). GPT-4 showed the highest performance, but even the smallest model, Llama-2 7B, demonstrated a comparable performance. Llama-2 7B is one of the smaller large language models among the widely used open-source ones.
>
>
> $\\phantom{X}$
>
> **Response to "Some baseline models are missing, such as GCC[1] and TCC[2]"**
>
> Thank you for the pointers. We have added them to the related work section in the revised manuscript.

---

### Official Review · Reviewer_t6rL · 2023-10-23

**Soundness:** 4 excellent
**Presentation:** 3 good
**Contribution:** 3 good
**Rating:** 8
**Confidence:** 4

**Summary:**

This paper proposes a new clustering paradigm that aims at clustering images based on the criterion provided by users. By leveraging the pre-trained VLM and LLM, the proposed method could understand the user's request and partition images accordingly.

**Strengths:**

1. I think the problem this work tackles is very interesting. By using the pre-trained VLM and LLM, the proposed paradigm could cluster the images based on the user's criterion, which favors real-world applications.
2. The proposed method improves the clustering explainability by providing the cluster names.
3. The effectiveness of the method is proved by experiments on both user-specified clustering and classic semantic clustering.

**Weaknesses:**

1. The paper is more like an engineering guidance instance of a research work. In other words, the proposed method "violently" clusters the images, all based on pre-trained VLM and LLM. Though I acknowledge the good question and paradigm raised by this work, I feel it is better to incorporate some clustering techniques into the paradigm (maybe in future works).
2. The refinement operation also seems a bit violent. It seems that if the user is not satisfied with the current clustering results, the whole process is simply repeated. I wonder if the refinement could integrate the experience learned from the last process.
3. 100 samples from 10 location classes might have biases by random sampling. I suggest enlarging the datasets of location and mood so that they would become reliable benchmarks for future studies.
4. What is the difference between sending the image description in step 1 and the main object in step 2 into step 3?
5. There is a related recent work Image Clustering with External Guidance (arXiv 2023) that also leverages the text modality to enhance image clustering, which the authors are encouraged to include in the related works.

**Questions:**

Please refer to the weaknesses.

---

> ### Author Response · Authors · 2023-11-21
> **Individual response to Reviewer t6rL**
>
> We thank the reviewer for the positive and constructive feedback. We are particularly excited to hear that the reviewer recognizes the problem we are tackling to be interesting and our experimental results convincing.
>
>
> $\\phantom{X}$
>
> **Response to "The paper is more like an engineering guidance instance of a research work. In other words, the proposed method "violently" clusters the images, all based on pre-trained VLM and LLM. Though I acknowledge the good question and paradigm raised by this work, I feel it is better to incorporate some clustering techniques into the paradigm (maybe in future works)."**
>
> The reviewer makes a great suggestion, namely incorporating existing clustering techniques into the IC|TC paradigm. This is something we had explored, and one approach that somewhat worked was using IC|TC as an initialization scheme for k-means.
>
> Specifically, we view the output of IC|TC as an initial clustering assignment, similar to the smart initial clustering assignment of k-means++. Then, we update the clusters through steps inspired by k-means, as more precisely described by the pseudo-code in the subsequent post. In our experiment with Stanford 40 Action, the clusters stabilized (converged) after 4 iterations. However, we did not observe a significant performance improvement to justify incorporating the result into the paper. Nevertheless, combining existing clustering techniques with IC|TC is indeed an interesting direction of future work that we plan to pursue.
>
>
> $\\phantom{X}$
>
> **Response to "The refinement operation also seems a bit violent. It seems that if the user is not satisfied with the current clustering results, the whole process is simply repeated. I wonder if the refinement could integrate the experience learned from the last process."**
>
> In the refinement process, it is not necessary that all three steps be repeated from scratch. In the Fair clustering experiment of section 4.4, we examined the results of Steps 1 through 3 and determined that it was enough to repeat Step 3 in the refinements.
>
>
> $\\phantom{X}$
>
> **Response to "100 samples from 10 location classes might have biases by random sampling. I suggest enlarging the datasets of location and mood so that they would become reliable benchmarks for future studies."**
>
> The reviewer raises a good point. We have additionally labeled 1000 images for the 'location' and 'mood' clustering tasks for the Stanford 40 action dataset and will release the label data so that it can be used as benchmarks for future studies. Based on these additional labels, we updated the evaluation numbers, and we found them to be similar to the prior numbers that we obtained with fewer labels.
>
>
>
>
> $\\phantom{X}$
>
> **Response to "What is the difference between sending the image description in step 1 and the main object in step 2 into step 3?"**
>
> The reviewer raises a great question, and this is indeed something we investigated ourselves. As an example, let us suppose that the image contains a singing man and an applauding audience and that we are using the text criterion 'action'. We may safely assume that, due to the expressive output from the VLMs, the output of step 1 will contain all information related to "singing" and "applauding." However, the output of step 2a, the raw label, may only capture "singing". Now, suppose that after step 2b, where the cluster names are obtained from the raw labels, there is a cluster name relating to 'applauding' but none directly related to 'singing'. Then, it is necessary for the LLM to be provided with the full textual description, not just the raw label, to properly assign the image to the cluster 'applauding'. In the initial stage of our work, we tried both methods and empirically observed that providing the output of Steps 1 to Step 3 showed superior performance compared to providing the output of Step 2a to Step 3.
>
>
>
>
> $\\phantom{X}$
>
>
> **Response to "There is a related recent work Image Clustering with External Guidance (arXiv 2023) that also leverages the text modality to enhance image clustering, which the authors are encouraged to include in the related works."**
>
> Thank you for this pointer. We have incorporated the discussion of this concurrent work into our revised manuscript.

---

> > ### Author Response · Authors · 2023-11-21
> > **Incorporating IC|TC and K-means**
> >
> > Our algorithm can be further enhanced by incorporating the algorithm design of k-means algorithm. One can view the output of our algorithm as an initial clustering assignment, similar to the smart initial clustering assignemnt of k-means++. Now we consider two representation spaces of the image. The first is the image description obtained from VLM (step 1), and the second is the raw label for the description obtained from LLM (step 2a). In generally, description is more high dimensional than raw label. So we will denote $Z_{high}$ be set of descriptions and $Z_{low}$ be set of raw labels.
> >
> > $\\phantom{X}$
> >
> > ### **Algorithm: Incorporating IC|TC and K-means**
> >
> > **Input:** Image descriptions $Z_{\text{high}}$, Raw labels $Z_{\text{low}}$, Number of clusters $K$,  Previous cluster assignment $C_{\text{prev}} \leftarrow \text{empty}$,  Current cluster assignment $C_{\text{curr}} \leftarrow \text{empty}$
> > **Output:** $C_{\text{curr}}$ and $\mu_{1:K}$
> >
> > 1. $\mu_{1:K} \leftarrow \text{Output of step 2b}$ &nbsp;
> >    `// initialize centroids through step 2b`
> >
> > 2. **Repeat**
> > 3.    &nbsp; &nbsp; **For** each point $z_{\text{high}} \in Z_{\text{high}}$
> > 4. &nbsp; &nbsp; &nbsp; &nbsp; $C_{\text{curr}}.\text{append}(\text{LLM}(\text{``Assign} \  z_{\text{high}} \ \text{to} \ \text{proper categories} \ \mu_{1:K} \text{''}))$ &nbsp; `//$C_{curr}$ is obtained through step 3`
> > 5. &nbsp; &nbsp; **If** $C_{\text{curr}} = C_{\text{prev}}$
> > 6. &nbsp; &nbsp; &nbsp; &nbsp; **break**
> > 7.   &nbsp; &nbsp; **Else**
> > 8. &nbsp; &nbsp; &nbsp; &nbsp;  $C_{\text{prev}} \leftarrow C_{\text{curr}}$
> > 9.  &nbsp; &nbsp; **For** $j = 1 \ \text{to} \ K$
> > 10. &nbsp; &nbsp; &nbsp; &nbsp;  $\mu_j \leftarrow \text{LLM}(\text{summarize} \ \{z_{\text{low}} \in c^{-1}(j)\} \ \text{to one category})$
> > 11. &nbsp; &nbsp;  **Until** $C_{\text{curr}} = C_{\text{prev}}$   `//until cluster assignments do not change`

---

> ### Comment · Reviewer_t6rL · 2023-11-22
> **Thanks for the response**
>
> I have a few concerns after reading the authors' response:
>
> Q3: Clustering results on 700 samples of PPMI (K=7) based on location also need to be reported.
>
> Q4: The corresponding discussions need to be added to the manuscript.
>
> Q5: Sorry but I did not see the mentioned discussion.

---

> > ### Author Response · Authors · 2023-11-22
> > **Re: Thanks for the response**
> >
> > Q3: We have carried out the labeling of the 700 images and updated the numbers.
> >
> > Q4: We have added the corresponding discussion to Appendix A.6 of the updated manuscript. (We present and discuss a real instance, rather than a hypothetical, which has a similar but slightly different description compared to our previous explanation.)
> >
> > Q5: We sincerely apologize for this mistake and the confusion it may have caused. As we were adding several references to the updated manuscript, we missed this one but mistakenly thought that we had processed it. Our updated manuscript now does include a discussion of the mentioned reference. Thank you for checking this meticulously and pointing it out.

---

> > > ### Comment · Reviewer_t6rL · 2023-11-22
> > > **Thanks for the response**
> > >
> > > Thanks for the response. My concerns have now been addressed.

---

### Official Review · Reviewer_zuGS · 2023-10-31

**Soundness:** 3 good
**Presentation:** 3 good
**Contribution:** 3 good
**Rating:** 6
**Confidence:** 5

**Summary:**

The authors propose an image clustering technique based on text criteria driven by LLMs. They correctly point out that unsupervised clustering often does not yield results that would satisfy a human. They therefore propose clustering based on user specified criteria, which they ask the user to provide and then build on with prompt engineering. They use a three step process to first get the user criteria from the user, then extract relevant textual features from the images and then cluster based on those textual features.

**Strengths:**

1. Good literature survey.
2. Interesting proposed algorithm for clustering.
3. Interesting results.

**Weaknesses:**

1. I am not convinced that this problem is not merely a form of retrieval based on user specifications.
2. The approach is inherently not testable at scale unless the authors expend a huge amount of human resources.
3. That is probably why the authors choose relatively small datasets of images that have strong structures.
4. I would be more favorably inclined if the authors could show that the proposed method is scalable to millions of images as per the state of the art.

**Questions:**

Please see my comments on weaknesses above.
I am satisfied with the authors' response to the extent that their method is scalable. I am therefore raising my score.

---

> ### Author Response · Authors · 2023-11-21
> **Individual response to Reviewer zuGS**
>
> We are happy to hear that Review zuGS finds our algorithm and results interesting. We positively address the reviewer's concerns in the following, including running a large-scale experiment to demonstrate that our approach is indeed scalable.
>
>
> $\\phantom{X}$
>
>
> **Response to "I am not convinced that this problem is not merely a form of retrieval based on user specifications."**
>
> The task of clustering requires (i) finding the clusters and (ii) cluster assignment. While image retrieval techniques can certainly be used for cluster assignment, finding the clusters is, in our view, not a form of retrieval. We discuss this point further in our individual response to Reviewer 32Bv.
>
>
> $\\phantom{X}$
>
>
>
> **Concerns about scalability**
>
> We clarify that our method *is* scalable. In principle, a simple back-of-the-envelope calculation shows that the algorithmic complexity of IC|TC is essentially linear in the number of images. To show that IC|TC is scalable in practice, we conducted an experiment with a quarter-million images during the rebuttal period. (Scaling to a million images, as suggested by the reviewer, would be possible if we had just a few more days to run the experiments.) Specifically, we used the Places dataset [20] and used 50 subclasses, each with 5,000 images, to create a quarter-million dataset. With the text criterion 'place', we achieve 70.5\% accuracy, which is competitive with the state-of-the-art image classification results. (There is some nuance to this comparison that we discuss in the main writing.) We detail the experimental setting and the results in Appendix C of our revised manuscript. We hope this quarter-million experiment alleviates the reviewer's concerns about scalability.
>
> [20] B. Zhou, A. Lapedriza, J. Xiao, A. Torralba, and A. Oliva. Learning deep features for scene recognition using places database. 2014.
>
> $\\phantom{X}$
>
> **Response to "The approach is inherently not testable at scale unless the authors expend a huge amount of human resources."**
>
> Evaluating/testing clustering and most unsupervised learning methods at scale is an inherent challenge to the task, but it is possible to carry out an approximate evaluation by sub-sampling a few data points from each cluster and obtaining human labels for those samples. In this sense, an approximate but fair evaluation IC|TC or any comparable image clustering method at scale does not require an inordinate amount of human resources.
>
> We also clarify that the text-criterion refinement of section 3.4 is an optional step and that it does not require a huge amount of human effort. It simply requires that the user roughly examine the clustering results to determine if they are satisfied; in fact, examining the results and fine-tuning hyperparameters is something a user of any clustering method will do in practice.
>
>
> $\\phantom{X}$
>
> **Response to "That is probably why the authors choose relatively small datasets ... I would be more favorably inclined if the authors could show that the proposed method is scalable to millions of images ...."**
>
> We hope our additional large-scale experiment addresses the reviewer's concern.
>
>
> $\\phantom{X}$
>
> We thank the reviewer for the constructive feedback. We believe we have addressed the reviewer's main concern about scalability. Since we do show that our proposed method is scalable to a quarter-million images, we kindly ask the reviewer to consider raising the score.

---

### Official Review · Reviewer_32Bv · 2023-11-03

**Soundness:** 3 good
**Presentation:** 3 good
**Contribution:** 2 fair
**Rating:** 6
**Confidence:** 5

**Summary:**

The paper describes a simple prompting method to perform image clustering using existing vision-language models and LLMs. The method is very straightforward, simply relying on image captioning, then prompting to distill keywords and perform clustering. There is little novelty, but the method performs well and the experiments demonstrate strong performance against recent baselines.

**Strengths:**

Interactive editing of the criterion to refine clustering results is a significant strength of the approach. This feature is practical, and intuitive for users, as is the basic premise of the user providing an initial clustering criterion.

The experiments are reasonable, and show that the method can incorporate semantic conditions on the clustering operation (which is done through a prompt clause). A variety of datasets are used to show different aspects of the work, and for comparison against baseline clustering methods that cannot incorporate conditioning. One particularly interesting experiment is to instruct the LLM to ignore gender, to mitigate bias, with successful results.

**Weaknesses:**

The comparison to related work does not include the body of work in language-guided image retrieval, which seems highly relevant here, particularly since many of those methods use a vision-language encoder and indexing scheme that clusters the image archive according to linguistic concepts, as the proposed approach does implicitly. However, given that the method does not address the underlying representation, but just relies on the LLM to perform clustering as a black box, this is a minor concern.

The discussion of prior work in deep clustering is also very brief, and does not position the proposed contributions against current, relevant work.

The method requires prior specification of the number of clusters, which is a significant limitation and drawback.

The method is very simple, just a set of straightforward text prompts. The crucial step of doing the clustering is just deferred to the LLM itself, by asking it to cluster N labels into K categories.

**Questions:**

I have revised my review after reading the other reviews and the authors' responses. I'm considerably more favorable on the paper, given its generally positive reception by others. The method does not present a clustering algorithm per se, and is more of an experimental analysis of how well an LLM can do clustering innately. However the experiments are sufficiently thorough and appears to be of interest to this community.

---

> ### Author Response · Authors · 2023-11-21
> **Individual response to Reviewer 32Bv**
>
> Reviewer 32Bv raises a valid criticism regarding missing comparisons with the image retrieval literature. We acknowledge that this is an important omission on our part caused by our lack of publication experience in the area of image retrieval, so we provide a thorough literature survey on image retrieval techniques in our revised manuscript. However, we disagree with the other points of criticism, and we feel that the reviewer is being overly harsh in the assessment.
>
> $\\phantom{X}$
>
> **Comparison with language-guided image retrieval.**
>
> The task of clustering is a combination of two sub-tasks: Finding the clusters and assigning individual data points to the clusters. (The standard K-means algorithm is literally an alternation of solving these two sub-tasks.) On the other hand, language-guided image retrieval is the process of searching for an image in a database that contains reference features from a given source image and language. Although these tasks may be related, they are certainly not the same; image retrieval techniques are very relevant to the sub-task of cluster assignment but not to the sub-task of finding the clusters. As a concrete example, we may cluster images based on the musical instrument being played, and IC|TC can automatically find the categories "brass instrument" and "string instrument" when we specify K=2 clusters. Image retrieval techniques can certainly retrieve images based on the precise instrument or the instrument category, but image retrieval, by itself, cannot determine that  "brass instrument" and "string instrument" are the best K=2 clusters for the specific dataset at hand.
>
>
> $\\phantom{X}$
>
> **Response to "The discussion of prior work in deep clustering is also very brief, and does not position the proposed contributions against current, relevant work."**
>
> We do not think the blanket statement that our discussion is "very (overly) brief" is accurate. Our discussion of prior work on deep clustering is split across Section 2 *'Comparison with classical clustering'* and Section 5 *'Related work'*, and we do position our proposed contributions against the relevant current state-of-the-art methods. (Incidentally, we, the authors, do have publication experience in clustering methods, unlike in image retrieval.) If the reviewer feels that relevant references in deep clustering have been overlooked, please provide specific pointers, and we will incorporate them into our discussion.
>
> $\\phantom{X}$
>
>
> **Response to "The method requires prior specification of the number of clusters, which is a significant limitation and drawback.'"**
>
> Most clustering algorithms, including the state-of-the-art methods, require the user to specify target cluster numbers. Therefore, we do not consider this feature to be a limitation or a drawback when positioned against current, relevant work.
>
>
>
> $\\phantom{X}$
>
> **Response to "The method is very simple, just a set of straightforward text prompts. The crucial step of doing the clustering is just deferred to the LLM itself, by asking it to cluster N labels into K categories."**
>
> In our view, the simplicity and straightforwardness of the method is a *strength*, not a weakness. Indeed, our contribution is a reframing of the clustering task in a way that defers the algorithmic heavy lifting to the VLM and LLM. However, finding and exploiting reductions (reframing a certain problem as an instance of another problem for which powerful tools exist) is a classical approach in computer science and machine learning, so it is unclear to us why the reviewer considers this reduction-based approach a weakness.
>
> The reviewer seems to agree that the task of clustering with text criterion is useful. We ask the reviewer to consider whether the experimental results (rather than the novelty of the method) are convincing and of interest to the ICLR community.

---

### Author Response · Authors · 2023-11-21
**Common response**

We sincerely thank the reviewers for their insightful comments and suggestions. We are happy to hear that Reviewers t6rL and heiR consider our newly proposed clustering paradigm very interesting and practically useful. Reviewer zuGS expressed concerns regarding scalability, which we positively address in our individual response.

Reviewers 32Bv and zuGS raise a very valid point regarding missing references to and comparisons with the existing image retrieval literature. We fully agree that it is important for the scientific process to present contributions within the proper context of the current literature. To remedy this omission, we wrote a thorough survey of the relevant prior work, attached it here in the rebuttal, and included it in our revised manuscript. However, our review of the image retrieval literature indicates that our present contribution does not conflict with the prior work on image retrieval. We detail this point in our individual response to Reviewer 32Bv.


Image clustering is a classical and commonly used machine learning task, and our contribution is a reframing of the task in a way that utilizes the recent foundation models. We ask the reviewers to consider whether the results we present are of interest to those in the ICLR community who use image clustering.


$\\phantom{X}$


Our revised manuscript includes the experiment on a large-scale dataset of the size of a quarter-million (Appendix C).

---

> ### Author Response · Authors · 2023-11-21
> **Image retrieval survey**
>
> Image retrieval aims to find images from a database that are relevant to a given query. This crucially differs from clustering in that clustering requires both finding the clusters and assigning the images to them; image retrieval techniques are very relevant to the sub-task of cluster assignment but not to the sub-task of finding the clusters.
>
> The fundamental approach in image retrieval is to assess the similarity among image features. Current approaches focus on two kinds of image representations: global features and local features. For global representations, [1, 2, 3, 4, 5]extracts activations from deep CNNs and aggregates them for obtaining global features. For local representations, [6, 7, 8, 9, 10, 11, 12] proposed well-embedded representations for all regions of interest. Recent state-of-the-art methods [7, 13, 4, 14, 15] typically followed a two-stage paradigm: initially, candidates are retrieved using global features, and then they are re-ranked with local features. Recently, [16, 17, 18, 19] proposed to condition retrieval on user-specified language.
>
>
> [1] A. Babenko, A. Slesarev, A. Chigorin, and V. Lempitsky. Neural codes for image retrieval. European
> Conference on Computer Vision, 2014.
>
> [2] G. Tolias, R. Sicre, and H. Jegou. Particular object retrieval with integral max-pooling of cnn ´
> activations. arXiv: 1511.05879, 2015.
>
> [3] A. Gordo, J. Almazan, J. Revaud, and D. Larlus. Deep image retrieval: Learning global representa- ´
> tions for image search. European Conference on Computer Vision, 2016.
>
> [4] B. Cao, A. Araujo, and J. Sim. Unifying deep local and global features for image search. European
> Conference on Computer Vision, 2020a.
>
> [5] S. Lee, S. Lee, H. Seong, and E. Kim. Revisiting self-similarity: Structural embedding for image
> retrieval. Conference on Computer Vision and Pattern Recognition, 2023.
>
> [6] K. M. Yi, E. Trulls, V. Lepetit, and P. Fua. Lift: Learned invariant feature transform. European
> Conference on Computer Vision, pages 467–483, 2016.
>
> [7] H. Noh, A. Araujo, J. Sim, T. Weyand, and B. Han. Large-scale image retrieval with attentive deep
> local features. International Conference on Computer Vision, 2017.
>
> [8] D. P. Vassileios Balntas, Edgar Riba and K. Mikolajczyk. Learning local feature descriptors with
> triplets and shallow convolutional neural networks. Proceedings of the British Machine Vision
> Conference (BMVC), 2016.
>
> [9] D. DeTone, T. Malisiewicz, and A. Rabinovich. Superpoint: Self-supervised interest point detection
> and description. Computer Vision and Pattern Recognition Workshops, 2018.
>
> [10] K. He, Y. Lu, and S. Sclaroff. Local descriptors optimized for average precision. Computer Vision
> and Pattern Recognition, 2018.
>
> [11] M. Dusmanu, I. Rocco, T. Pajdla, M. Pollefeys, J. Sivic, A. Torii, and T. Sattler. D2-net: A trainable cnn for joint description and detection of local features. Conference on Computer Vision and
> Pattern Recognition, 2019.
>
> [12] J. Revaud, C. De Souza, M. Humenberger, and P. Weinzaepfel. R2d2: Reliable and repeatable
> detector and descriptor. Neural Information Processing Systems, 2019.
>
> [13] O. Simeoni, Y. Avrithis, and O. Chum. Local features and visual words emerge in activations.
> Conference on Computer Vision and Pattern Recognition, 2019.
>
> [14] Z. Zhang, L. Wang, L. Zhou, and P. Koniusz. Learning spatial-context-aware global visual feature
> representation for instance image retrieval. International Conference on Computer Vision, 2023.
>
> [15] H. Wu, M. Wang, W. Zhou, Z. Lu, and H. Li. Asymmetric feature fusion for image retrieval. 2023.
>
> [16] N. Vo, L. Jiang, C. Sun, K. Murphy, L.-J. Li, L. Fei-Fei, and J. Hays. Composing text and image for
> image retrieval - an empirical odyssey. Computer Vision and Pattern Recognition, 2019.
>
> [17] Z. Liu, C. Rodriguez-Opazo, D. Teney, and S. Gould. Image retrieval on real-life images with
> pre-trained vision-and-language models. International Conference on Computer Vision, 2021.
>
> [18] A. Baldrati, M. Bertini, T. Uricchio, and A. Del Bimbo. Conditioned and composed image retrieval
> combining and partially fine-tuning clip-based features. Conference on Computer Vision and
> Pattern Recognition, 2022.
>
> [19] Y. Tian, S. Newsam, and K. Boakye. Fashion image retrieval with text feedback by additive attention
> compositional learning. Conference on Applications of Computer Vision, 2023.

---

### Meta-Review · Area_Chair_VPYs · 2023-12-05

**Metareview:**

While there were some concerns, most reviewers comment that they have been addressed by the rebuttal (or those reviewers had positive scores to begin with). While the contribution seems more experimental, there is sufficient support for this contribution that justifies acceptance.

**Justification For Why Not Higher Score:**

Not a lot of confident high scores

**Justification For Why Not Lower Score:**

No scores below 6

---

### Decision · Program_Chairs · 2024-01-16

Accept (poster)